# NEURAL AUTO-DESIGNER FOR ENHANCED QUANTUM KERNELS

**Cong Lei**[1]    **Yuxuan Du**[2,†]    **Peng Mi**[1]    **Jun Yu**[3]    **Tongliang Liu**[1,†]

[1] Sydney AI Centre, School of Computer Science, The University of Sydney    [2] JD Explore Academy
[3] Department of Automation, University of Science and Technology of China
†: corresponding authors (duyuxuan123@gmail.com, tongliang.liu@sydney.edu.au)

## ABSTRACT

Quantum kernels hold great promise for offering computational advantages over classical learners, with the effectiveness of these kernels closely tied to the design of the quantum feature map. However, the challenge of designing effective quantum feature maps for real-world datasets, particularly in the absence of sufficient prior information, remains a significant obstacle. In this study, we present a data-driven approach that automates the design of problem-specific quantum feature maps. Our approach leverages feature-selection techniques to handle high-dimensional data on near-term quantum machines with limited qubits, and incorporates a deep neural predictor to efficiently evaluate the performance of various candidate quantum kernels. Through extensive numerical simulations on different datasets, we demonstrate the superiority of our proposal over prior methods, especially for the capability of eliminating the kernel concentration issue and identifying the feature map with prediction advantages. Our work not only unlocks the potential of quantum kernels for enhancing real-world tasks but also highlights the substantial role of deep learning in advancing quantum machine learning.

## 1 INTRODUCTION

Quantum computing presents a compelling prospect for revolutionizing machine learning, harnessing the unique attributes of quantum mechanics such as superposition and entanglement (Feynman, 2018; Biamonte et al., 2017). While many studies have showcased the ability of quantum computers to reach quadratic or even exponential runtime improvements for solving linear equations and matrix factorizations (Harrow et al., 2009; Rebentrost et al., 2014; Lloyd et al., 2014; Harrow & Montanaro, 2017; Du et al., 2018; Gilyén et al., 2019; Du et al., 2022a), their practical realization for real-world applications poses significant challenges primarily due to the demanding error-corrected qubit requirements (Babbush et al., 2021; Gouzien & Sangouard, 2021). Presently, quantum hardware has entered the noisy intermediate-scale quantum (NISQ) era (Preskill, 2018), characterized by limited qubit counts, shallow circuit depth, inherent system noise, and constrained topologies. Consequently, great efforts have been directed towards exploring quantum machine learning (QML) algorithms that can effectively operate on NISQ machines (Cerezo et al., 2021; Bharti et al., 2022). Among these efforts, quantum neural networks (Mitarai et al., 2018; Abbas et al., 2021; Du et al., 2021; 2022c; Tian et al., 2023; Qian et al., 2022) and quantum kernels (Havlíček et al., 2019; Schuld & Killoran, 2019; Blank et al., 2020) stand out as leading proposals. Celebrated by the flexibility of quantum kernels, proof-of-principle experiments have been conducted to exhibit their feasibility in solving tasks in various domains (Wu et al., 2021; Haug et al., 2021; Park et al., 2020; Krunic et al., 2022).

The experimental advance spurs the exploration of computational advantages of quantum kernels when executed on NISQ machines. In pursuing this inquiry, a recent theoretical study pointed out that quantum advantages may arise if two conditions are met (Kübler et al., 2021): the reproducing kernel Hilbert space formed by quantum kernels encompasses functions that are computationally hard to evaluate classically, and the target concept resides within this class of functions. According to the construction rule of quantum kernels, this means that their power tightly hinges on the quantum feature map, typically composed of input-dependent gates to encode classical data into the quantum state feature space and trainable quantum gates with a specified layout. This statement has been

verified when quantum kernels are applied to tackle well-structured tasks, encompassing classification of synthetic datasets (Huang et al., 2021), discrete logarithmic problems (Liu et al., 2021), quantum phase recognition (Wu et al., 2023b), and specific decision problems (Jäger & Krems, 2023).

Different from well-structured problems, identifying quantum feature maps that satisfy both two conditions concurrently poses a significant challenge for most realistic datasets. As a result, quantum kernels associated with inappropriate quantum feature maps tend to be inferior to that of classical kernels (Huang et al., 2021). Even worse, when the quantum feature map involves deep circuit depth, a large number of qubits, and too many noisy gates, the corresponding quantum kernel may encounter the vanishing similarity (Thanasilp et al., 2023a) and the degraded generalization ability (Wang et al., 2021), precluding any potential computational advantages.

To address this issue, several initial approaches have been explored, including the design of problem-specific quantum feature maps (Hubregtsen et al., 2022), rescaling the bandwidth of input data (Shaydulin & Wild, 2022; Canatar et al., 2023), and mitigating the negative impact of system noise and shot error (Wang et al., 2021; Shastry et al., 2022). These approaches are complementary to one another and aim to enhance the performance of quantum kernels. Specifically, in the realm of designing problem-specific quantum feature maps, two primary streams of research have emerged: tuning variational parameters within the quantum feature maps (Hubregtsen et al., 2022; Glick, 2022; Gentinetta et al., 2023) and employing evolutionary or Bayesian algorithms to continually adjust the arrangement of quantum gates within the feature map (Altares-López et al., 2021; Pellow-Jarman et al., 2023; Torabian & Krems, 2023). However, both methods encounter severe caveats. The former overlooks the influence of quantum gate layout (see Table 1 for supporting evidence), while the latter requests high computational overhead solely by adjusting the layout. Besides, none of them addresses the vanishing similarity issue encountered when processing high-dimensional data and struggles to adapt to the limited topology of NISQ machines. Given these considerations, a crucial question arises: *How can we design a generic method to efficiently construct problem-specific quantum feature maps capable of (i) adapting to high-dimensional data into modern quantum machines without encountering the vanishing similarity issue; (ii) accommodating both layout and parameter optimization with satisfactory performance?*

Table 1: **Evidence for the effect of gate layout in performance of quantum kernels**. Test accuracies of 1000 quantum kernels on tailored MNIST dataset whose feature map consists of the different gate layouts.

| max acc: 82.67% | min acc: 44.67 % | avg acc: 68.70 % | std: 8.23% |
|---|---|---|---|

In this work, we utilize advanced deep-learning techniques to overcome the limitations aforementioned. Specifically, we frame the quantum kernel design as a *discrete-continuous joint optimization* problem and then propose a data-driven method dubbed quantum kernel design by neural networks (QuKerNet) to solve this optimization problem, which amounts to effectively and automatically designing problem-specific quantum feature maps. To our best knowledge, this is the first framework that enables the automatic design of kernels by simultaneously considering both circuit layouts and variational parameters. Besides, QuKerNet is more resource-friendly to modern quantum hardware as it takes into account the topology of qubits and the available number of qubits. Despite that neural architecture search and quantum architecture search also orient to the discrete-continuous joint optimization, adapting these proposals to design quantum kernels is not straightforward, because of the expensive computational costs of quantum kernels compared to (quantum) neural networks in collecting accurate predictions of training data. In this regard, we devise a surrogate loss to efficiently optimize QuKerNet.

Two core components of QuKerNet are the feature-selection technique and a neural predictor. The incorporation of the feature-selection technique enables QuKerNet to handle high-dimensional data and overcome the limitations posed by near-term quantum machines with limited qubits, thereby ensuring the practical utility and suppressing the vanishing similarity issue. Moreover, the exploitation of a neural predictor allows QuKerNet to distill knowledge from different quantum kernels on a given dataset, guaranteeing efficient and accurate performance prediction for the novel quantum feature map. Extensive numerical simulations are conducted to validate the efficacy of QuKerNet.

**The key motivation of our work**. 1) Quantum kernel has a rich theoretical foundation. 2) Quantum kernel is not equivalent to the Quantum Neural Network (QNN). 3) Quantum Architecture Search (QAS) for quantum kernel search is not straightforward.

On the one hand, quantum kernel design is quite different from the field of deep learning, where there are many well-performing structures that can serve as basic modules to construct neural networks, such as CNN, RNN, and Transformer. In contrast, quantum machine learning is still in its early stages, and we are uncertain about which well-structured modules can be used to build quantum kernels. This situation is even more challenging in the near-term quantum devices, as it requires considering noise, topology, and various limitations. Designing a good quantum kernel becomes more challenging.

Therefore, designing better quantum kernels is both urgent and crucial, as it represents a key step towards achieving potential quantum advantage on near-term quantum devices.

## 2 PRELIMINARIES

### 2.1 QUANTUM COMPUTATION

Quantum mechanics works in the Hilbert space $\mathcal{H}$ with $\mathcal{H} \cong \mathbb{C}$, where $\mathbb{C}$ represents the complex Euclidean space. We use Dirac notation to denote quantum states. A *pure quantum state* is defined by a vector $|\cdot\rangle$ (named 'ket') with the unit length. The mathematical expression of a state is $|\boldsymbol{a}\rangle = \sum_{i=1}^{d} \boldsymbol{a}_i \boldsymbol{e}_i = \sum_{i=1}^{d} \boldsymbol{a}_i |i\rangle \in \mathbb{C}^d$ with $\sum_i |\boldsymbol{a}_i|^2 = 1$, where the computational basis $|i\rangle$ stands for the unit basis vector $\boldsymbol{e}_i \in \mathbb{C}^d$. The inner product of two quantum states $|\boldsymbol{a}\rangle$ and $|\boldsymbol{b}\rangle$ is denoted by $\langle\boldsymbol{a}|\boldsymbol{b}\rangle$ or $\langle\boldsymbol{a}, \boldsymbol{b}\rangle$, where $\langle\boldsymbol{a}|$ refers to the conjugate transpose of $|\boldsymbol{a}\rangle$. A state $|\boldsymbol{a}\rangle$ is in *superposition* if the number of nonzero entries in $\boldsymbol{a}$ is larger than one. Analogous to the 'ket' notation, *density operators* can be used to describe more general quantum states. Given a mixture of $m$ quantum pure states $|\psi_i\rangle \in \mathbb{C}^d$ with the probability $p_i$ and $\sum_{i=1}^{m} p_i = 1$, the density operator $\rho$ presents the mixed state $\{p_i, |\psi_i\rangle\}_{i=1}^{m}$ as $\rho = \sum_{i=1}^{m} p_i \rho_i$ with $\rho_i = |\psi_i\rangle\langle\psi_i| \in \mathbb{C}^{d \times d}$ and $\text{Tr}(\rho) = 1$.

The basic element in quantum computation is the quantum bit (*qubit*). A qubit is a two-dimensional quantum state that can be written as $|\boldsymbol{a}\rangle = \boldsymbol{a}_1 |0\rangle + \boldsymbol{a}_2 |1\rangle$. Let $|\boldsymbol{b}\rangle$ be another qubit. The quantum state represented by these two qubits is formulated by the tensor product, i.e., $|\boldsymbol{a}\rangle \otimes |\boldsymbol{b}\rangle$ as a 4-dimensional vector. Following conventions, $|\boldsymbol{a}\rangle \otimes |\boldsymbol{b}\rangle$ can also be written as $|\boldsymbol{a}, \boldsymbol{b}\rangle$ or $|\boldsymbol{a}\rangle |\boldsymbol{b}\rangle$. For clearness, we sometimes denote $|\boldsymbol{a}\rangle |\boldsymbol{b}\rangle$ as $|\boldsymbol{a}\rangle_A |\boldsymbol{b}\rangle_B$, which means that the qubits $|\boldsymbol{a}\rangle_A (|\boldsymbol{b}\rangle_B)$ is assigned in the quantum register $A$ $(B)$. There are two typical quantum operations. The first one is *quantum (logic) gates* operating on a small number qubits. Any quantum gate corresponds to a unitary transformation and can be stated in the circuit model, e.g., an $N$-qubit quantum gate $U \in SU(2^N)$ satisfies $UU^\dagger = U^\dagger U = \mathbb{I}_{2^N}$. The second one is the *quantum measurement*, aiming to extract quantum information such as the computation result into the classical form. Given a density operator $\rho$, the outcome $m$ will be measured with the probability $p_m = \text{Tr}(\mathbf{K}_m \rho \mathbf{K}_m^\dagger)$ and the post-measurement state will be $\mathbf{K}_m \rho \mathbf{K}_m^\dagger / p_m$ with $\sum_b \mathbf{K}_b^\dagger \mathbf{K}_b = \mathbb{I}$.

Any unitary can be decomposed into a set of basis gates. The basis gate set explored in this study is $\mathcal{G} = \{H, R_X(\alpha), R_Y(\beta), R_Z(\gamma), \text{CNOT}\}$, containing Hadamard gate, three rotational single-qubit gates along $X, Y$ and $Z$-axis, respectively, and one two-qubit Control-not gate. The mathematical expression of these five basis gates is $H \equiv \frac{1}{\sqrt{2}}\begin{bmatrix} 1 & 1 \\ 1 & -1 \end{bmatrix}$, $R_\sigma(\theta) = \exp(-i\theta\sigma/2)$ with $\sigma \in \{X, Y, Z\}$, $X \equiv \begin{bmatrix} 0 & 1 \\ 1 & 0 \end{bmatrix}, Y \equiv \begin{bmatrix} 0 & -i \\ i & 0 \end{bmatrix}, Z \equiv \begin{bmatrix} 1 & 0 \\ 0 & -1 \end{bmatrix}$, and $\theta \in [0, 2\pi]$, and $\text{CNOT} \equiv |0\rangle\langle0| \otimes \mathbb{I}_2 + |1\rangle\langle1| \otimes \text{X}$.

### 2.2 MECHANISM OF QUANTUM KERNELS

Kernel methods provide a powerful framework to perform nonlinear and nonparametric learning, attributed to their universal property and interpretability. Suppose that both the training and test examples are sampled from the same domain $\mathcal{X} \times \mathcal{Y}$. The training dataset is denoted by $\mathcal{D} = \{\boldsymbol{x}^{(i)}, y^{(i)}\}_{i=1}^{n} \subset \mathcal{X} \times \mathcal{Y}$, where $\boldsymbol{x}^{(i)} \in \mathbb{R}^d$ and $y^{(i)} \in \mathbb{R}$ refer to the $i$-th example with the feature dimension $d$ and the corresponding label, respectively. A general construction rule of kernel methods is embedding the given input $\boldsymbol{x}^{(i)} \in \mathbb{R}^d$ into a high-dimensional feature space, i.e., $\phi(\cdot) : \mathbb{R}^d \to \mathbb{R}^q$ with $q \gg d$, which allows that different classes of data points can be readily separable. Considering that explicitly manipulating $\phi(\boldsymbol{x}^{(i)})$ becomes computationally expensive for large $q$, kernel methods construct a kernel matrix $\text{K} \in \mathbb{R}^{n \times n}$ to effectively accomplish the learning tasks in the feature space. Specifically, the elements of K represent the inner product of feature maps with $\text{K}_{ij} = \text{K}_{ji} = \langle\phi(\boldsymbol{x}^{(i)}), \phi(\boldsymbol{x}^{(j)})\rangle$ for $\forall i, j \in [n]$, where such an inner product can be evaluated

by a positive definite function $\kappa(\boldsymbol{x}^{(i)}, \boldsymbol{x}^{(j)})$ in $O(d)$ runtime. The aim of kernel learning is using the employed kernel $K \in \mathbb{R}^{n \times n}$ to infer a hypothesis $h(\boldsymbol{x}^{(i)}) = \langle \boldsymbol{\omega}^*, \phi(\boldsymbol{x}^{(i)}) \rangle$ with a high test accuracy, where $\boldsymbol{\omega}^*$ are optimal parameters minimizing the loss function

$$\mathcal{L}(\boldsymbol{\omega}) = \lambda \langle \boldsymbol{\omega}, \boldsymbol{\omega} \rangle + \sum_{i=1}^{n} (\langle \boldsymbol{\omega}, \phi(\boldsymbol{x}^{(i)}) \rangle - y^{(i)})^2. \tag{1}$$

The performance of kernel methods heavily depends on the utilized embedding function $\phi(\cdot)$, or equivalently $\kappa(\cdot, \cdot)$. As such, various kernels such as the radial basis function kernel, Gaussian kernel, circular kernel, polynomial kernel, and *quantum kernel* have been proposed to tackle various tasks.

The mechanism of quantum kernels differs from the classical kernels in designing the feature map $\phi(\cdot)$. For an $N$-qubit quantum kernel, the prepared quantum state for the $i$-th example yields $|\varphi(\boldsymbol{x}^{(i)}, \boldsymbol{\theta})\rangle = U_E(\boldsymbol{x}^{(i)}, \boldsymbol{\theta}) |0\rangle^{\otimes N}$, where $U_E(\cdot, \boldsymbol{\theta})$ is the specified encoding quantum circuit with trainable parameters $\boldsymbol{\theta} \in \Theta$ and $\Theta$ being the parameter space. Denote $\rho(\boldsymbol{x}^{(i)}, \boldsymbol{\theta}) = |\varphi(\boldsymbol{x}^{(i)}, \boldsymbol{\theta})\rangle \langle \varphi(\boldsymbol{x}^{(i)}, \boldsymbol{\theta})|$. The aim of quantum kernel learning is seeking a quantum kernel $\mathrm{W}(\boldsymbol{x}, \boldsymbol{\theta}) \in \mathbb{R}^{n \times n}$ with $\boldsymbol{\theta} \in \Theta$, i.e.,

$$\mathrm{W}_{ij}(\boldsymbol{x}, \boldsymbol{\theta}) = \mathrm{Tr}(\rho(\boldsymbol{x}^{(i)}, \boldsymbol{\theta}) \rho(\boldsymbol{x}^{(j)}, \boldsymbol{\theta})), \ \forall i, j \in [n], \tag{2}$$

to infer a hypothesis formulated in Eq. (1) subjecting to a higher test accuracies than other kernels. In this regard, the power of quantum kernels is dominated by the exploited $U_E(\cdot, \boldsymbol{\theta})$. Notably, the implementation of $U_E(\cdot, \boldsymbol{\theta})$ is flexible and diverse. For an $N$-qubit quantum computer with the basis gate set $\mathcal{G}$, the generic form of the encoding circuit is

$$U_E(\boldsymbol{x}, \boldsymbol{\theta}) = \prod_{j=1}^{L} U_j(\boldsymbol{\alpha}_j) V_j \in SU(2^N), \ \forall j \in [L], \tag{3}$$

where $U_j(\boldsymbol{\alpha}_j)$ refers to any parameterized gate in $\mathcal{G}$ (e.g., $R_X$, $R_Y$ and $R_Z$ gates) acting on an arbitrary qubit, $\boldsymbol{\alpha}_j$ is an element in the set $\{\boldsymbol{x}, \boldsymbol{\theta}\}$, and $V_j \in \{\mathrm{CNOT}, \mathbb{I}_2\}$ refers to the fixed gate in $\mathcal{G}$ acting on arbitrary two qubits, and $L$ represents the total number of quantum gates.

## 3  QUKERNET: AUTOMATIC NEURAL DESIGNER FOR QUANTUM KERNELS

Let us first revisit the optimization perspective of quantum feature map design to highlight the shortcomings of previous approaches before moving to elaborate our proposal. As stated in Sec. 2.2, the performance of quantum kernels heavily depends on the exploited quantum feature map $|\varphi(\boldsymbol{x}, \boldsymbol{\theta})\rangle$ in Eq. (2), which has enormous choices attributed to the diversity of $\Theta$ and the gate arrangement of $\{(U_j, V_j)\}$ in Eq. (3). Taking account into such diversity, the loss function of quantum kernel learning in Eq. (1) should be reformulated as

$$\min_{\boldsymbol{\omega}, S \in \mathcal{S}, \boldsymbol{\theta} \in \Theta} \mathcal{L}(\boldsymbol{\omega}, S, \boldsymbol{\theta}) = \lambda \langle \boldsymbol{\omega}, \boldsymbol{\omega} \rangle + \sum_{i=1}^{n} (\langle \boldsymbol{\omega}, \varphi_S(\boldsymbol{x}^{(i)}, \boldsymbol{\theta}) \rangle - y^{(i)})^2, \tag{4}$$

where $|\varphi_S(\boldsymbol{x}^{(i)}, \boldsymbol{\theta})\rangle = U_E(\boldsymbol{x}^{(i)}, \boldsymbol{\theta}; S) |0\rangle^{\otimes N}$ and $U_E(\cdot, \cdot; S)$ amounts that the gate arrangement $S \in \mathcal{S}$ is used to build the encoding circuit. Note that locating the global optima $(S^*, \boldsymbol{\theta}^*)$ to minimize $\mathcal{L}(\boldsymbol{\omega}, S, \boldsymbol{\theta})$ is difficult, caused by the two aspects: (i) the loss landscape with respect to $\boldsymbol{\theta}$ is non-convex; (2) the number of feasible gate arrangements $|\mathcal{S}|$ exponentially scales with $N$ and $|\mathcal{G}|$.

Towards the optimization difficulty, initial attempts have been made to seek sub-optimal solutions of Eq. (4) in an efficient manner. Concretely, Altares-López et al. (2021) discarded the parameter space $\boldsymbol{\theta}$ and only focuses on searching the optimal gate arrangement $S \in \mathcal{S}$ to form $U_E$; Lloyd et al. (2020); Hubregtsen et al. (2022) adopted the fixed circuit layout $S$ and focus on optimizing $\boldsymbol{\theta}$. Unlike prior studies, our proposal *jointly optimizes the gate layout and parameters to approach the global minima of $\mathcal{L}$ in Eq. (4)*. In this point of view, all the attempts aforementioned are special cases of our method.

Another aspect overlooked in prior literature is the design of effective quantum kernels for high-dimensional datasets, which presents even greater challenges compared to the low-dimensional case. That is, an improper selection of gate arrangements can lead to the vanishing similarity issue, also known as kernel concentration (Huang et al., 2021; Thanasilp et al., 2023a) in which the off-diagonal elements of matrix $\mathrm{W}$ progressively diminish as the number of qubits $N$ and the total number of quantum gates $L$ increase. Consequently, quantum kernels exhibit lower test accuracy compared to

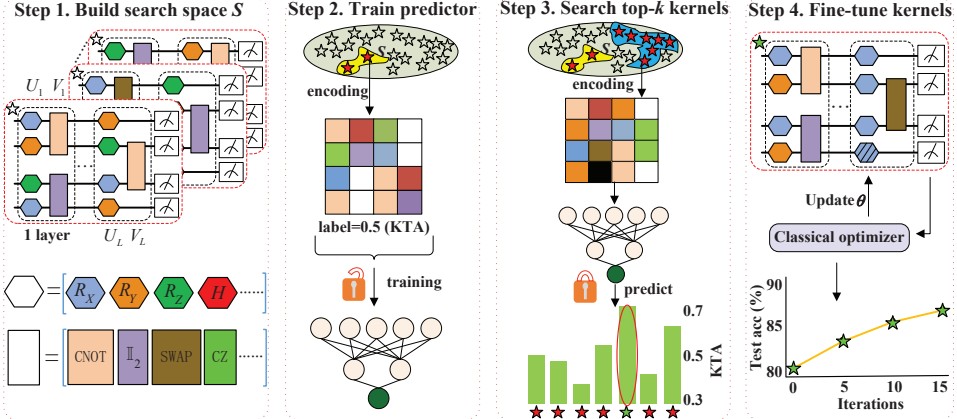

Figure 1: **The workflow of QuKerNet.** In step 1, QuKerNet sets up the search space $\mathcal{S}$ via the accessible basic quantum gate set $\mathcal{G}$. For example, the set of single-qubit gates includes $R_X$, $R_Y$, $R_Z$ represented by colored hexagons, and the set of two-qubit gates contains CNOT, CZ, SWAP represented by the colored rectangles. In step 2, a pentagram represents a feasible candidate circuit in the search space $\mathcal{S}$. In the training process, $M$ candidate circuits (highlighted by the red pentagrams) are collected and transformed to an image via the proposed encoding method. Meanwhile, KTA of these sampled candidate circuits is calculated according to Eq. (5). These training data are used to train an MLP-based neural predictor. In step 3, the optimized neural predictor is employed to predict the performance of a large number of candidate circuits sampled from the search space $\mathcal{S}$. Afterward, QuKerNet ranks the predicted KTA and selects the top $k$ candidate circuits. In step 4, for each candidate circuit, QuKerNet randomly replaces $m$ parameterized gates, which are used to encode data features, by the variational parameters $\boldsymbol{\theta}$, highlighted by the hexagon with shadow. These parameters are optimized to maximize KTA. Then we compare their classification accuracy and the circuit with the highest accuracy is selected as the output circuit.

other kernels. Additionally, modern quantum machines with limited and noisy quantum resources make it challenging to directly embed high-dimensional data features into the encoding circuit. This is because when an extremely large value of $L$ is used to encode the data, executing such quantum kernels on NISQ machines introduces a significant amount of errors, resulting in degraded performance (Wang et al., 2021).

## 3.1 IMPLEMENTATION OF QUKERNET

Here we devise a data-driven approach to automatically design the near-optimal quantum kernel for the specified learning task by minimizing $\mathcal{L}$ in Eq. (4). Our approach, dubbed quantum kernel design by neural networks (QuKerNet), consists of four steps: search space setup, neural-predictor training, top-$k$ quantum kernels search, and fine-tune. An intuition is depicted in Fig. 1. Conceptually, QuKerNet completes the optimization of Eq. (4) via a two-stage learning strategy. In the first stage, QuKerNet estimates the optimal circuit layout $S^* \in \mathcal{S}$, accomplished by the first three steps. In the second stage, QuKerNet estimates the optimal $\boldsymbol{\theta} \in \Theta$ under the searched circuit layout, as completed by the last step. Through decoupling the discrete optimization from the continuous optimization, QuKerNet enables an efficient and scalable way to automatically design enhanced quantum kernels. In the remainder of this subsection, we introduce the implementation of each step and defer the omitted details to Appendix B.

*Search space setup.* This step aims to design a suitable search space $\mathcal{S}$ in Eq. (4), determined by $N$ and $L$. Concretely, for a large $N$ and $L$, most of the candidate circuit layouts in $\mathcal{S}$ yield a poor performance, due to the vanishing similarity issue. To this end, QuKerNet applies the feature selection (FS) method, i.e., Max-Relevance and Min-Redundancy (mRMR) (Peng et al., 2005), to the dataset in the preprocessing step, where a dimension-reduction function $g : \mathbb{R}^d \to \mathbb{R}^p$ operates with each example $\boldsymbol{x}^{(i)}$ to extract $p$ entries from $d$ entries and form the new example $\hat{\boldsymbol{x}}^{(i)}$ with $p \ll d$. This approach not only mitigates the vanishing similarity issue but also facilitates the manipulation of high-dimensional data on NISQ machines. Once the new dataset $\hat{\mathcal{D}} = \{\hat{\boldsymbol{x}}^{(i)}, y^{(i)}\}_{i=1}^n$ is prepared, QuKerNet establishes $\mathcal{S}$ with the constraint $NL \sim O(p)$.

*Neural-predictor training.* This step targets to train a neural predictor to capture the relation between $S \in \mathcal{S}$ and the performance of the corresponding quantum kernel. Specifically, the neural predictor takes a set of circuit layouts in $\mathcal{S}$ as input and predicts their training accuracy on the dataset $\hat{\mathcal{D}}$. The optimization of neural predictor follows the supervised learning paradigm, where the weights of the neural network are optimized to minimize the discrepancy between its predictions and the true training accuracies via gradient descent methods. Through this training process, the neural predictor is expected to provide accurate predictions for the performance of unseen circuit layouts in $\mathcal{S}$.

Nevertheless, unlike QAS, the evaluation of the training accuracy of a quantum kernel with a specified circuit is computationally expensive. Therefore, an alternative strategy is needed to efficiently collect a labeled dataset $\mathcal{T}$ for training neural predictors. In our approach, we replace the training accuracy with the kernel-target alignment (KTA) as the label that the neural predictor predicts (Cristianini et al., 2001) since KTA is a reliable surrogate for classification accuracy and can be effectively calculated. Given a circuit layout $S$ and the preprocessed dataset $\hat{\mathcal{D}}$, its KTA yields

$$\mathsf{K}(\hat{\mathcal{D}}; S) = \frac{1}{\mathcal{N}} \sum_{ij} y^{(i)} y^{(j)} \, \mathsf{W}_{i,j}(\hat{\boldsymbol{x}}, \boldsymbol{\theta}; S), \tag{5}$$

where $\mathsf{W}_{i,j}(\hat{\boldsymbol{x}}, \boldsymbol{\theta}; S)$ refers to the $(i, j)$-th entry of the quantum kernel $\mathsf{W}$ in Eq. (2) whose circuit layout is $S$, $y^{(i)} \in \{0, 1\}$ is the binary label of the $i$-th example, and $\mathcal{N} = n \|\mathsf{W}\|_F$ denotes the normalization factor. For multi-class datasets with $R$ classes, its KTA is computed by replacing $y^{(i)} y^{(j)}$ with $J_{i,j}$, i.e, $J_{i,j} = 1$ if $y^{(i)} = y^{(j)}$; otherwise, $J_{i,j} = -1/(R-1)$ (Camargo & González, 2009). Refer to Appendix C for the comparison of the runtime cost between KTA and the direct optimization. To facilitate calculation, we set $\Theta = \emptyset$ so that all parameterized quantum gates in $S$ encode the data feature without the trainable parameters $\boldsymbol{\theta}$. By calculating KTA of $M \sim O(poly(p))$ different circuit layouts from $\mathcal{S}$ in parallel, we obtain the labeled dataset $\mathcal{T} = \{S^{(i)}, \mathsf{K}(\hat{\mathcal{D}}; S^{(i)})\}_{i=1}^M$.

The neural predictor utilized in QuKerNet comprises three essential components: input vectorization, model implementation, and optimization strategy. The input vectorization step addresses the conversion of the unstructured format of the circuit layout $S$ into a structured representation that can be processed by deep neural networks. In this regard, we employ the transformation strategy proposed by (Zhang et al., 2021) to accomplish this task. For the implementation of the neural predictor, we adopt the Multilayer Perceptron as the underlying architecture (Goodfellow et al., 2016). The specific configuration of the neural network, including the number of hidden layers, is determined based on the characteristics of the datasets being considered.

*Top-k quantum kernels search.* The purpose of this step is to select the most promising quantum kernels in the search space. Firstly, we randomly sample $M'$ circuit layouts $\mathcal{S}'$ from $\mathcal{S}$ with $\mathcal{S}' \subset \mathcal{S}$ and $|\mathcal{S}'| = M'$. Then, the trained neural predictor is used to predict the $\mathsf{K}(\hat{\mathcal{D}}; S)$ with $S \in \mathcal{S}'$. Afterward, these $M'$ circuits are sorted according to $\mathsf{K}(\hat{\mathcal{D}}; S)$ and the top $k$ circuit layouts are preserved to construct the candidate layout set $\mathcal{S}_c$.

*Fine-tune.* This step aims to improve the performance of quantum kernels in $\mathcal{S}_c$. To do so, given $S \in \mathcal{S}_c$, we randomly reset $m$ encoding gates in $U_E(\boldsymbol{x})$ as tunable parameters, i.e., $U_E(\boldsymbol{x}; S)$ turns to be $U_E(\boldsymbol{x}, \boldsymbol{\theta}; S)$ and $\Theta \neq \emptyset$. Besides, to ensure the performance of quantum kernels, the parameters $\boldsymbol{\theta}$ are initialized using the average value of corresponding features in the relevant gates. Then we fine-tune the quantum kernels to maximize the KTA with the fixed $S$ by updating $\boldsymbol{\theta}$. After training, we choose the circuit layout with the highest classification accuracy from $K$ fine-tuned candidates as the searched quantum kernel.

Then the quantum kernel $\mathsf{W}^*$ can be obtained based on the $(S^*, \boldsymbol{\theta}^*)$. Consequently, the optimal $\boldsymbol{\omega}^*$ can be achieved through $\boldsymbol{\omega}^* = \sum_{i=1}^n \sum_{j=1}^n \phi(\boldsymbol{x}^{(i)})((\mathsf{W}^* + \lambda I)^{-1})_{ij} y^{(j)}$, which can be used for kernel learning.

### 3.2 Variants of QuKerNet

Our proposed scheme in QuKerNet offers a general framework that can be seamlessly integrated with various advanced methods and techniques. For instance, the gate set $\mathcal{G}$ used to construct the search space can be modified to accommodate different quantum hardware platforms, enabling flexibility and adaptability. Furthermore, different feature selection methods such as mRMR (Peng et al., 2005), Principal Component Analysis (PCA) (Abdi & Williams, 2010) and Locality Preserving Projections

(LPP) (He & Niyogi, 2003) and other methods (Lei & Zhu, 2018; Yuan et al., 2021; Li et al., 2018; Zhu et al., 2018; 2019) can be employed to handle diverse high-dimensional datasets effectively. In terms of the neural predictor, alternative backbones and encoding methods can be explored to enhance its performance in capturing the relationship between circuit layouts and kernel performance, i.e., graph neural network (GNN) backbone plus graph encoding strategy (He et al., 2023b), Convolutional Neural Network (CNN) backbone with image encoding (Zhang et al., 2021). Additionally, advanced training methods can be employed to optimize the neural predictor with improved performance, i.e., data augmentation (Jaitly & Hinton, 2013; Huang et al., 2023b), Dropout (Srivastava et al., 2014), early stopping (Caruana et al., 2000). Moreover, the fine-tuning process can benefit from the incorporation of more sophisticated heuristic methods, enhancing the overall optimization process.

## 4  NUMERICAL RESULTS

In this section, we conduct a set of experiments to evaluate the performance of QuKerNet across various datasets. Our primary objectives are to explore the potential merits of the quantum kernel designed by QuKerNet compared to prior quantum kernels and classical kernels, assess the effectiveness of QuKerNet in addressing the vanishing similarity issue for large-scale quantum kernels, and evaluate the robustness of QuKerNet in noisy environments. Refer to Appendix B and C for the elaboration of model implementations, datasets, and more simulation results.

**Datasets**. We benchmark QuKerNet on three datasets, each representing a distinct domain: computer vision, finance, and learning tasks relevant to the demonstration of quantum advantages. The first dataset utilized is a tailored version of the MNIST dataset (Lecun & Bottou, 1998), consisting of handwritten digit images. To ensure computational efficiency, we distill 300 images from the original MNIST dataset, focusing on the labels from 0 to 4, and reduce the feature dimension of each example from 784 to 40 using mRMR. The second dataset employed is a tailored Credit Card (CC) dataset (Dal Pozzolo et al., 2017), commonly used for fraud detection and risk assessment in the financial industry. For this dataset, we extracted 200 (100 for each class) samples from the CC dataset and reduce the dimension of each example from 28 to 24 using PCA. Last, we utilize the method proposed in (Huang et al., 2021) to relabel the tailored MNIST dataset (class 0 and 1), creating a synthetic dataset that allows us to evaluate the potential advantages of QuKerNet in a controlled manner.

**Source code and hardware**. All simulations are conducted using Python, utilizing the PennyLane (Bergholm et al., 2018), PyTorch (Paszke et al., 2019), and the JAX library (Bradbury et al., 2018). All experiments are run on AMD EPYC 7302 16-Core Processor (3.0GHz) with 188G memory (Ubuntu system). The source code for our implementation will be made publicly available on GitHub repository. https://github.com/tmllab/2024_ICLR_QuKerNet.

**Implementation of QuKerNet**. Throughout the entire experiment, we utilize the hardware efficient ansatz (HEA) (Kandala et al., 2017) as the backbone to construct the search space of QuKerNet. In particular, as shown in Fig. 2(a), HEA subsumes a block-wise layout, and each block is composed of parameterized single-qubit gates and fixed two-qubit gates. In this context, the encoding circuit $U_E(\boldsymbol{x}, \boldsymbol{\theta}) = \prod_{j=1}^{L} U_j(\boldsymbol{\alpha}_j)V_j = \prod_{j'=1}^{B} \hat{U}_{j'}(\boldsymbol{\beta})$, where $\hat{U}_{j'}(\boldsymbol{\beta})$ refers to the $j'$-th block taking the form as $\hat{U}_{j'} = (\otimes_{i=1}^{N} R_a(\boldsymbol{\beta}_{ij'}))\hat{V}$, $a \in \{X, Y, Z\}$, $\boldsymbol{\beta}_{ij'}$ is an element in the set $\{\boldsymbol{x}, \boldsymbol{\theta}\}$, and $\hat{V}$ refers to the entangled layer consisting of CNOT gate and $\mathbb{I}_2$ gate. The construction rule of the search space is as follows. For the parameterized single-qubit gates in each block, when the index of qubits $i$ satisfies $i \pmod 2 = 0$, we apply the same parameterized gate $R_a$ (each gate in $R_a$ appears at a probability of 1/3); otherwise, we adopt another type of parameterized gate $R_{a'}$. For the entangled gates in each block, when the index of qubit $i$ satisfies $i < N$, we pick a two-qubit gate from $\{\text{CNOT}, \mathbb{I}_2\}$ uniformly random and apply it to the $i$-th and $(i+1)$-th qubits. This strategy not only facilitates the balance of the size of the search space and the expressivity but also enables an efficient implementation of the circuit on real quantum devices. With a slight abuse of notations, in the remained section, we denote $L_0$ as the layer number, contrasting with the original definition of gate number in Eq. (3). For instance, when the $U_E$ encodes all the features of $\boldsymbol{x}$, the layer number of $U_E$ is seen as 1 and denoted by $L_0 = 1$ and the relationship between $L_0$ and the number of blocks $B$, is $B = L_0 * \lceil p/N \rceil$.

**Other hyper-parameter settings**. Here we only introduce the general hyper-parameter settings and defer the specific hyper-parameter settings to the corresponding experiments. Except for the KTA

Figure 2: **Role of feature selection of QuKerNet.** (a) The depiction of hardware efficient ansatz (HEA). Each layer is composed of $R_X$ gates where the rotational angle on the $i$-th qubit amounts to the $i$-th feature of the input data, and CNOT gates acting on the adjacent qubits. (b) Alleviation of vanishing similarity by feature selection. The label of "KV" represents kernel variance. Both "F" and 'dimensions' refer to the feature dimension. (c) Comparison with two feature selection methods, i.e., mRMR and random selection.

experiment, $N = 8, L_0 \in \{1, 2, 3, 4, 5\}, M' = 50000, k = 10, |\Theta| = 20$. And the neural predictor is optimized by Adam with 0.01 learning rate for 30 epochs, and the criterion used is Smooth L1 Loss. Each setting is repeated with 5 times to collect the statistical results.

**Performance metrics**. We use test accuracy (Acc) and the kernel variance (KV) to evaluate the performance of QuKerNet, which quantifies the generalization ability and inspects the degree of vanishing similarity of the specified kernel, respectively. The mathematical expression of KV for the kernel W is $\mathrm{Var}(\mathrm{W}) = \mathbb{E}(\mathrm{W}^2) - (\mathbb{E}(\mathrm{W}))^2$, where the expectation is taken over the input data $\boldsymbol{x}$.

In the following, we present our numerical simulation results to exhibit the effectiveness of QuKerNet.

**FS is necessary to alleviate the vanishing similarity issue**. Here we show how feature selection affects the performance of the tailored MNIST (with 150 training examples sampled from the distilled dataset). The focus on the tailored MNIST rather than the rest two datasets is because of its high-dimensionality feature. The results are visualized in Fig. 2(b). The left line chart indicates that when $L_0$ is fixed, as $p$ increases, KV continuously decreases. More precisely, when $L_0 = 5$, during the process of increasing $p$ from 8 to 40, KV decreases from approximately 0.05 to 0.001. The right line chart reveals that without feature selection, even if $L_0 = 1$, the KV is close to 0.0001. By contrast, after feature selection, it is around 0.001 even when $L_0 = 9$.

**FS versus random pick**. To further explore the role of feature selection, we searched for a 40-dimensional feature subset from the tailored MNIST using two methods (mRMR and random pick) to construct two new datasets. Additionally, we randomly generated 300 kernels with $L_0 \in \{1, 2, 3, 4, 5\}$ to conduct kernel learning on these two datasets. The box chart in Fig. 2(c) highlights that the data preprocessed by feature selection attains a higher classification accuracy. Taken together, it can be concluded that the feature selection technique can alleviate the vanishing similarity issue to a certain extent and improve the learning performance of quantum kernels.

**KTA is a good surrogate of training accuracy**. To avoid the expensive computational overhead, QuKerNet adopts KTA instead of training accuracy to construct the labeled dataset to train the neural predictor, as elucidated in Sec. 3.1. To verify the effectiveness of this replacement, we investigate the correlation between the performance of our proposal under the loss functions in Eq. (4) and Eq. (1) on the same tiny MNIST at two learning stages i.e., QuKerNet-1 (only optimize the circuit layout $S$) and QuKerNet-2 (simultaneously optimize the circuit layout $S$ and variational parameters $\boldsymbol{\theta}$), respectively. The tiny MNIST contains 50 train and 50 test examples, with labels ranging from 0 to 4. Each example consists of 4 features selected by mRMR. In this case, $N = 4, L_0 = 1, |\Theta| = 50, |\mathcal{S}| = 72$ and $\mathcal{S}$ contains in total $3 * 3 * 2^3 = 72$ different circuit layouts. The restricted search space allows us to explore the whole search space with a thorough analysis. The simulation results are exhibited in Fig. 3. In this case, we use the Pearson correlation coefficient

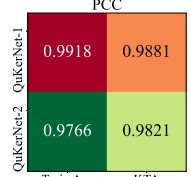

Figure 3: Correlation of kernels optimized under different loss functions.

(PCC, higher is better) to evaluate the validity of KTA. From Fig. 3, the PCCs between the training accuracy of kernels found by QuKerNet and the train accuracy of kernels selected by Eq. (1), are 0.9918 (QukerNet-1) and 0.9766 (QuKerNet-2), respectively. This indicates we can employ Eq. (4) instead of Eq. (1) to implement kernel learning. Besides, KTA has been demonstrated to be a reasonable surrogate as the PCCs between the KTA of kernels selected by QuKerNet and the training accuracy of kernels found through Eq. (1) are both close to 1 (i.e., 0.9881 in QukerNet-1 and 0.9821

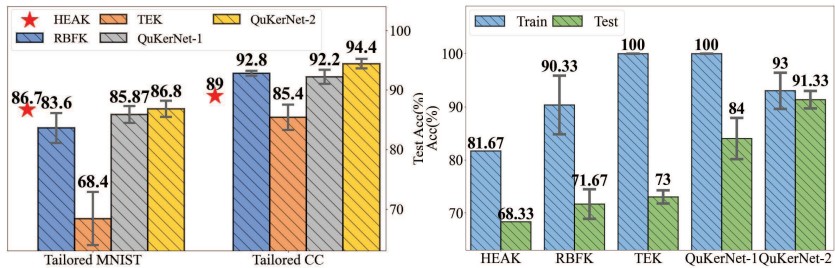

Figure 4: **Numerical results of QuKerNet.** (a) The best kernels discovered by TEK, QuKerNet-1, and QuKerNet-2 on tailored MNIST and tailored CC datasets. (b) The training and test accuracy for the best kernels searched by TEK, QuKerNet-1, and QuKerNet-2 on the synthetic dataset. Note that the results of HEAK have no standard deviation as it is a fixed kernel without any adjustable parameters.

in QuKerNet-2). These results validate the feasibility of using KTA instead of training accuracy to construct the labeled dataset to train the neural predictor.

**Enhanced performance after each step of QuKerNet.** We next exploit the performance of QuKerNet at different learning stages when learning the three datasets introduced in

Table 2: The top test accuracy for three datasets on different stages.

|  | Training data | Candidate | Fine-tune |
|---|---|---|---|
| Tailored MNIST | $82.13 \pm 0.27$ | $85.87 \pm 1.43$ | $\mathbf{86.80 \pm 1.36}$ |
| Tailored CC | $87.00 \pm 0.00$ | $92.20 \pm 1.17$ | $\mathbf{94.40 \pm 0.80}$ |
| Synthetic dataset | $80.33 \pm 0.67$ | $84.00 \pm 3.89$ | $\mathbf{91.33 \pm 1.63}$ |

'**Datasets**'. We set $M = 1000, 500, 1000$ for tailored MNIST, tailored CC, and synthetic dataset, respectively. As shown in Table 2, QuKerNet is able to find kernels that perform better than those in the training set, indicating that it has the ability to genuinely grasp certain rules to infer kernel performance based on the circuit layout. Furthermore, it can be observed that the top performance of the kernels selected by QuKerNet keeps improving in both QuKerNet-1 and QuKerNet-2, demonstrating that optimizing both $S$ and $\boldsymbol{\theta}$ is necessary and effective.

**Comparison with other kernels**. To further verify the performance of kernels found by QuKerNet is superior to unoptimized or individually optimized ($S$ or $\boldsymbol{\theta}$) algorithms, we compare the performance of five different methods, Radial Basis Function Kernels (RBFK) (Buhmann, 2000), HEA-based quantum kernel (HEAK) (Thanasilp et al., 2023b), Training Embedding Kernels (TEK) (Hubregtsen et al., 2022), QuKerNet-1, and QuKerNet-2 in learning three datasets aforementioned (see Appendix A.3 for more details about comparison algorithms). The statistical results of five random experiments conducted on tailored MNIST, tailored CC, and synthetic datasets are shown in Fig. 4. Compared to the other algorithms, QuKerNet-2 has achieved the highest test accuracy on three datasets (tailored MNIST: 86.8%, tailored CC: 94.4%, synthetic data: 91.33%), showcasing its superiority.

**Potential advantages.** We last apply HEAK, RBFK, TEK, QuKerNet-1, and QukerNet-2 to learn the synthetic dataset, with the purpose of exploring whether the kernels designed by QuKerNet may possess certain quantum advantages. From Fig. 4(b), it can be observed that there is a huge performance gap between TEK and the kernel designed by QuKerNet (i.e., 73% versus 91.33%), which demonstrates the importance of optimizing $S$. Besides, the RBFK achieves a very low test accuracy compared to quantum kernels, including QuKerNet (i.e., 71.67% versus 91.33%), implying the potential of QuKerNet compared to classical kernels. Moreover, the small gap between train and test accuracy hints good generalization ability of QuKerNet compared to other kernels. Furthermore, the results of QuKerNet-1 and QuKerNet-2 (16% versus 1.67%) suggest that optimizing $\boldsymbol{\theta}$ is crucial to improve the generalization of QuKerNet.

## 5 CONCLUSION

We propose QuKerNet to automatically design problem-specific quantum feature maps, which greatly improves the power of quantum kernels under NISQ settings. In contrast with prior advantageous quantum kernels established on well-structured problems, our proposal does not require prior information on tasks at hand. This characteristic underpins the potential of QuKerNet to use NISQ machines to conquer realistic tasks with computational merits.

ACKNOWLEDGMENTS

Jun Yu was supported by the Natural Science Foundation of China (62276242), National Aviation Science Foundation (2022Z071078001), CAAI-Huawei MindSpore Open Fund (CAAIXSJLJJ-2021-016B, CAAIXSJLJJ-2022-001A), Anhui Province Key Research and Development Program (202104a05020007), Dreams Foundation of Jianghuai Advance Technology Center (2023-ZM01Z001), USTC-IAT Application Sci. & Tech. Achievement Cultivation Program (JL06521001Y), Sci. & Tech. Innovation Special Zone (20-163-14-LZ-001-004-01). Tongliang Liu is partially supported by the following Australian Research Council projects: FT220100318, DP220102121, LP220100527, LP220200949, IC190100031. The authors would give special thanks to Xinbiao Wang from Wuhan University for helpful discussions.

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

The structure of our Appendix is as follows. Appendix A gives an introduction to the related work, noise quantum system, comparison algorithms we used as well as vanishing similarity of quantum kernels as preliminaries of our paper. Appendix B provides more details of our QuKerNet framework introduced in 3.1 of the main text. Appendix C provides more experimental details and results to help us better understand the capability of QuKerNet.

# A  PRELIMINARIES

This section serves as an introduction to provide essential background knowledge that will enhance our understanding of this paper. The key areas covered include the related work, the noise quantum system, the comparative algorithms employed in this study, and the phenomenon of vanishing similarity of quantum kernels.

## A.1  RELATED WORK

Prior literature most related to our work can be classified into two categories: quantum kernels and quantum architecture search (QAS). Here we separately review the connections and differences between these studies and our work.

**Kernel-based quantum algorithms.**  Previous studies of quantum kernels can be divided into two groups. The first group is data-independent quantum kernels (Blank et al., 2020; Huang et al., 2021; Thanasilp et al., 2023b; Huang et al., 2023a), which use a fixed circuit to embed the data. These quantum kernels may not have good prediction as the quantum feature map may be inappropriate for the manipulated dataset. Another group is data-dependent quantum kernels (Glick, 2022; Lloyd et al., 2020; Hubregtsen et al., 2022), where the quantum feature is customized to the given dataset. The approaches of designing data-dependent quantum features either focus on modifying the circuit layout or tuning variational parameters, indicating that these methods can only achieve local optimal solutions. QuKerNet differs from prior works in the pursuit of the global optima, by simultaneously considering the circuit layout and variational parameters. In this perspective, our work is a highly generalized version of prior kernel algorithms, where they are all special cases of our algorithm.

**QAS.**  QAS aims to automatically design the architecture of quantum neural networks that can attain high performance for a specified learning task. Essentially, QAS is a discrete-continuous joint optimization problem. Depending on the different optimization methods, QAS can be classified into heuristic-based QAS (Zhang & Zhao, 2022; Sünkel et al., 2023), reinforcement learning based QAS (Ostaszewski et al., 2021), Bayesian-based QAS (Duong et al., 2022), and other methods (He et al., 2023b; Zhang et al., 2021; He et al., 2023a; Zhang et al., 2022; Lu et al., 2023; Wu et al., 2023a; Du et al., 2022b; Linghu et al., 2022). Different from QAS, QuKerNet orients to enhance quantum kernels. The fundamental difference between neural networks and kernels hints the hardness of directly employing QAS to automatically design quantum kernels.

## A.2  NOISY QUANTUM SYSTEM

Besides Dirac notation, the density matrix can be used to describe more general qubit states evolved in open quantum systems. Formally, the density matrix of an $n$-qubit pure state $|\psi\rangle$ is $\rho = |\psi\rangle \langle\psi| \in \mathbb{C}^{2^n \times 2^n}$, where $\langle\psi| = |\psi\rangle^\dagger$ refers to the complex conjugate transpose of $|\psi\rangle$. For an ensemble of pure states of a quantum system, $\{p_j, |\psi_j\rangle\}_{j=1}^m$ with $p_j > 0$, where $|\psi_j\rangle$ is one state of this quantum system and $j$ is an index, $\sum_{j=1}^m p_j = 1$, and $|\psi_j\rangle \in \mathbb{C}^{2^n}$ for $j \in [m]$, its density matrix is $\rho = \sum_{j=1}^m p_j \rho_j$ with $\rho_j = |\psi_j\rangle \langle\psi_j|$ and $\text{Tr}(\rho) = 1$.

## A.3  MECHANISM OF CLASSICAL AND QUANTUM KERNELS

In the main text, we adopt three kernels, i.e., radial basis function kernel (RBFK), hardware efficient ansatz-based quantum kernel (HEAK), and training embedding quantum kernel (TEK), to benchmark the performance of our proposal. For completeness, here we provide the necessary backgrounds of three kernels.

**Radial basis function kernel**. RBFK is a common kernel used in machine learning. The mathematical form of RBFK is

$$\mathrm{K}_{ij} = \exp(-\gamma ||\boldsymbol{x}^{(i)} - \boldsymbol{x}^{(j)}||^2) \tag{6}$$

where $\gamma$ is a hyper-parameter. RBFK is commonly used in various domains such as image classification (Chapelle et al., 1999), face detection (Bartlett et al., 2003), text classification (Gao & Sun, 2010), protein structure prediction (Mandle et al., 2012), and many other areas (Jiang et al., 2018).

**Hardware efficient ansatz-based kernel**.

$$U_E(\boldsymbol{x}, \boldsymbol{\theta}) = \prod_{j=1}^{B} \hat{U}_j(\boldsymbol{\beta}) \tag{7}$$

where $\hat{U}_j(\boldsymbol{\beta})$ refers to the $j$-th block taking the form as $\hat{U}_j = (\otimes_{i=1}^{N} R_a(\boldsymbol{\beta}_{ij}))\hat{V}, a \in \{X, Y, Z\}, \boldsymbol{\beta}_{ij}$ is an element in the set $\{\boldsymbol{x}, \boldsymbol{\theta}\}$, and $\hat{V}$ refers to the entangled layer consisting of CNOT gate and $\mathbb{I}_2$ gate. And HEAK is widely used in the fields of quantum machine learning (Nakaji & Yamamoto, 2021), quantum chemistry (Kandala et al., 2017), and combinatorial optimization (Leone et al., 2022) due to its implementability and expressibility on NISQ devices.

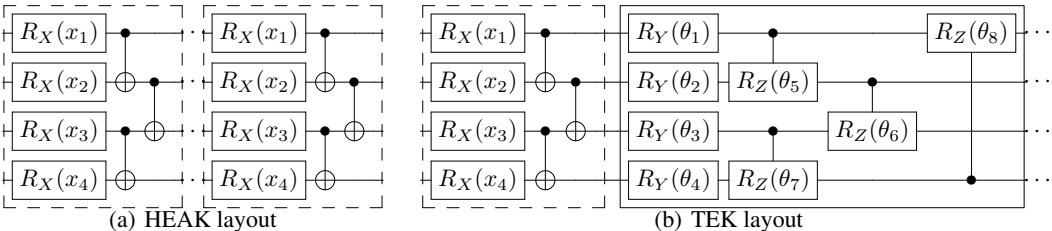

(a) HEAK layout      (b) TEK layout

Figure 5: **HEAK layout and TEK layout**. The $\boldsymbol{x}_j$ is corresponding to the $j$-th feature in sample $\hat{\boldsymbol{x}}^{(i)}$. (a) The layout of HEAK. And each dashed region is a block that is used to encode up to $N$ features of $\hat{\boldsymbol{x}}^{(i)}$. (b) The layout of TEK. The layout of the dashed region is used for encoding data, while the solid line portion represents variational parameters used to adjust the performance of the kernel. A trainable module is connected after each block of HEAK. The dashed area and the solid line area together form a block.

**Training embedding kernels**. TEK is a method that only optimizes the $\boldsymbol{\theta}$ for a given $S$. In this paper, TEK is designed based on HEAK, and their layouts are shown in Fig. 5. Therefore, it takes the following mathematical form:

$$U_E(\boldsymbol{x}, \boldsymbol{\theta}) = \prod_{j=1}^{B} \hat{U}_j(\boldsymbol{\beta})\widetilde{U}_j(\boldsymbol{\gamma}) \tag{8}$$

where $\hat{U}_j(\boldsymbol{\beta})$ is the same to the definition in Eq. (7). While $\widetilde{U}(\cdot)$ is a flexible layout used to adjust the quantum states of encoding, and $\boldsymbol{\gamma}$ represents the variational parameters. In this study, we use the circuit layout proposed in (Hubregtsen et al., 2022) to construct $\widetilde{U}(\cdot)$. So $\widetilde{U}_j(\boldsymbol{\gamma}) = (\otimes_{i=1}^{N} R_Y(\boldsymbol{\gamma}_{ij}))\widetilde{V}$, $\boldsymbol{\gamma}_{ij}$ is an entry of $\boldsymbol{\theta}$, and $\widetilde{V}$ refers to the entangled layer. And for the each index of qubit $i$, we apply the $CR_Z$ gate to the $i$-th and $(i+1) \pmod{N}$-th qubits. TEK is suitable for image classification (Lloyd et al., 2020) or regression tasks (Liu et al., 2022).

In summary, RBFK and HEAK are static kernels with deterministic feature mapping. While HEAK and TEK are quantum kernels and these kernels are built upon the same circuit layout $S$.

### A.4   VANISHING SIMILARITY OF QUANTUM KERNELS

The vanishing similarity of quantum kernels is that as the size of the problem increases, the difference between kernel values becomes increasingly small, which means that kernel values will tend towards the same value. More precisely, vanishing similarity can be formally defined as follows:

$$\mathrm{P}[|\mathrm{W}_{ij} - \mathbb{E}(\mathrm{W}_{ij})| \geq \delta] \leq \frac{\beta^2}{\delta^2}, \beta \in O(1/b^N) \tag{9}$$

where $b > 1$. If $\text{Var}[\text{W}_{ij}] \in O(1/b^n)$ and for all $\text{W}_{ij}$, Eq. (9) holds true, we identify that this is a phenomenon of vanishing similarity. In general, there are four main triggers that lead to vanishing similarity, including high expressibility, global measurements, entanglement and noise (Thanasilp et al., 2023a). When this phenomenon occurs, it becomes challenging to extract meaningful information from the quantum states in a reasonable amount of time. This is because distinguishing subtle differences between quantum states to evaluate quantum kernels requires an exponential number of measurements. Therefore, vanishing similarity leads to the poor performance of quantum kernels. To address this issue, one should be avoid to design a highly expressible, entangled quantum kernel. For the global measurements-induced vanishing similarity, one can employ problem-specific quantum kernel to remit it (Liu et al., 2021).

## B    IMPLEMENTATION DETAILS OF QUKERNET

In this section, we elaborate on the missing details of QuKerNet in the main text. In particular, we separately present the implementation details of each stage of QuKerNet, followed by the summarization of its Pseudocode.

**Search space setup**. We visualize the construction rule of search space in Fig. 6. Specifically, given the dataset $\hat{\mathcal{D}} = \{\hat{\boldsymbol{x}}^{(i)}, y^{(i)}\}_{i=1}^n$, there are several strategies to load the classical data into an $N$-qubit quantum system. Suppose that the feature dimension of $\hat{\boldsymbol{x}}^{(i)}$ is $p$. In the case of $N = p$, the first way is element-wise encoding, where the variational quantum gates operating with $j$-th qubit load the $j$-th feature of $\hat{\boldsymbol{x}}^{(i)}$, i.e., $\boldsymbol{\alpha}_j = \hat{\boldsymbol{x}}_j^{(i)}$ with $\boldsymbol{\alpha}_j$ is defined in Eq. (3). An alternative way is random encoding, where each variational quantum gate loads one non-repetitive feature of $\boldsymbol{x}^{(i)}$.

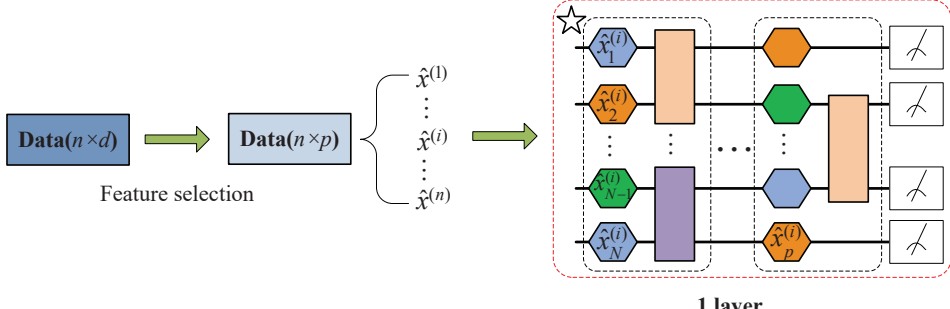

**1 layer**

Figure 6: **The step 1 of QuKerNet**. First, we preprocess the data through feature selection. Next, we design the search space for the circuits based on relevant rules. Then, we encode the preprocessed data using different circuits sampled from the search space.

We next consider the scenario with $p < N$ and design two encoding approaches. First, for the variational quantum gates acting on the $j$-th qubit with $j \in [N]$, the encoded data feature is $\hat{\boldsymbol{x}}_{j'}^{(i)}$ with $j' = j \pmod{p}$. The second approach involves random parameters. For the variational quantum gate operating with the first $p$ qubits, we set $\boldsymbol{\alpha}_j = \hat{\boldsymbol{x}}_j^{(i)}$ for $\forall j \in [p]$. Besides, for the rest $N - p$ qubits, the parameters of the variational quantum gates are randomly and uniformly sampled from the interval $[0, 2\pi)$.

Last, in the case of $p > N$, we design three encoding methods as exhibited in Fig. 7. The first way is sequential encoding, which encodes features sequentially. The second way is chain encoding, which utilizes the chain queue to load features. The third way is random encoding. In this case, the $\boldsymbol{\alpha}_j$ is set to the value of the feature randomly selected from $\hat{\boldsymbol{x}}^{(i)}$.

We note that there are multiple ways of using random encoding to form a multi-layer encoding layout. The first solution is replicating the entire layout of a single block multiple times. Another solution is expanding each dotted area in Fig. 8 multiple times. It is also feasible to repeat the layout of the individual dotted areas in Fig. 8 for the varied times.

**Neural-predictor training**. Neural-predictor adopted in QuKerNet is composed of two parts: image encoder and the deep neural networks. The former intends to convert the information circuit layout into a format that can be processed by neural networks. The latter is employed to distill the knowledge

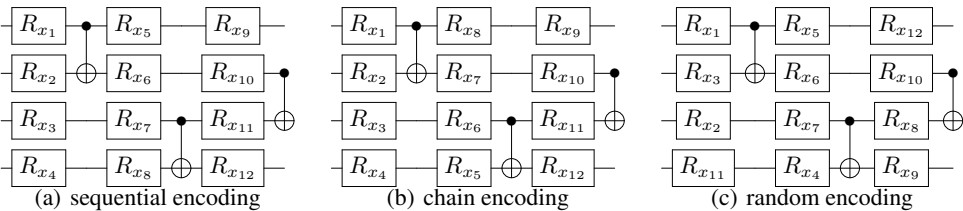

Figure 7: Various encoding methods. The $x_j$ is corresponding to the $j$-th feature in sample $\hat{\boldsymbol{x}}^{(i)}$.

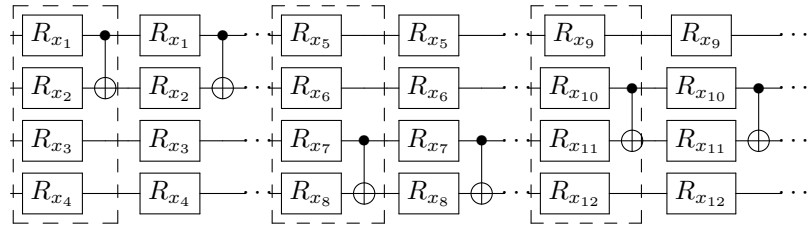

Figure 8: Repeated encoding scheme. The $x_j$ is corresponding to the $j$-th feature in sample $\hat{\boldsymbol{x}}^{(i)}$.

between circuit layout (i.e., quantum feature map) and the corresponding performance. In the following, we separately explain their implementations.

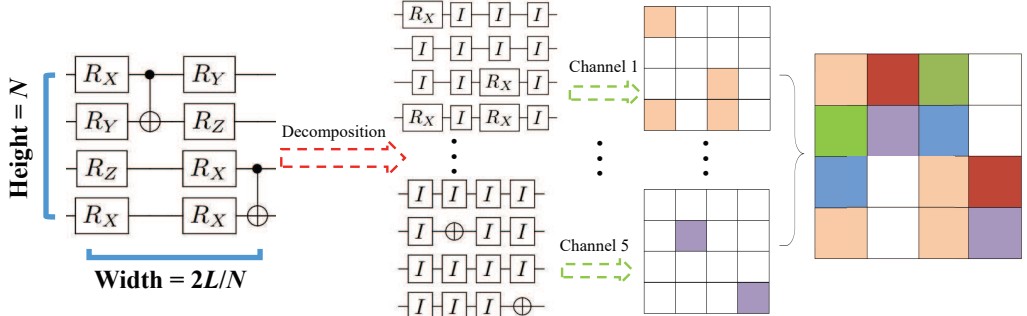

Figure 9: **Representation of quantum circuit layout**. The number of qubits $N = 4$, and the total number of rotation gates $L = 8$. First, the input circuit can be divided into five separate layouts, each layout containing only one type of gate. Then, these layouts are individually encoded into images, with each image corresponding to a channel. Finally, the images from the five channels are combined to form the final output image.

The implementation of the image encoder is shown in Fig. 9. In particular, we encode circuits into images based on the scheme proposed in (Zhang et al., 2021). For circuits with varied depth, we fill the encoded images with blank pixels to the maximum width of images in this set to unify the size of these images (refer to Fig. 10(a)). Specifically, $N$ is seen as the height of the image, and $2L/N$ corresponds to the width of the image. In addition, each type of gate associates with an image channel. Note that CNOT gate corresponds to 2 channels, because it includes a target qubit and a control qubit. And there are only two pixel values in each channel, that is, 0 and 1, which indicates whether a gate has been applied or not (1 represents a gate has been applied, 0 means there is either no gate or $\mathbb{I}_2$ gate has been applied on this position), and the position of the pixel is identical with the position of this gate in quantum circuit.

The implementation of the neural network adopted in QuKerNet is depicted in Fig. 11. That is, a simple Multilayer Perceptron (MLP) is employed as the regression model. This model is a 2-layer MLP model, including one hidden layer and one output layer. And the total number of trainable parameters of this model is $1280L + 257$, where $L$ refers to the total number of quantum rotation gates. Adam optimizer is applied in our experiments and the learning rate is set to 0.01, and the batch

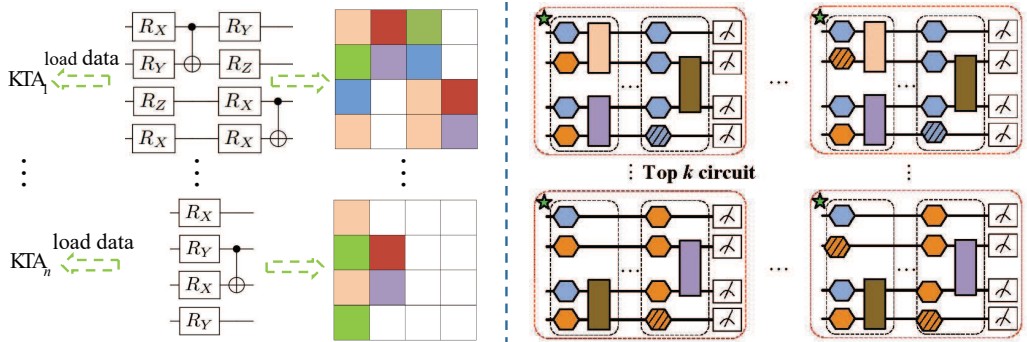

Figure 10: **Some details of step 2 and 4 in QuKerNet**. (a) The left area represents data collection for neural network training. First, all the circuits sampled from $\mathcal{S}$ will be encoded into images of a uniform size (with the width determined by the maximum depth among all circuits). Then, these circuits also are used to load the data and calculate the KTA using Eq. 5. Finally, the pairs of images and KTA can be used to train the neural network. (b) The right area is the fine-tuning stage. For each of the $k$ circuits found through the QuKerNet-1, the QuKerNet performs $|\boldsymbol{\theta}|$ different parameterized gate replacements. In each replacement, the number and positions of replaced gates remain the same for the $k$ circuits.

size is 32. Besides, in order to achieve better prediction results, we magnify the value of KTA 10 times in each data pair.

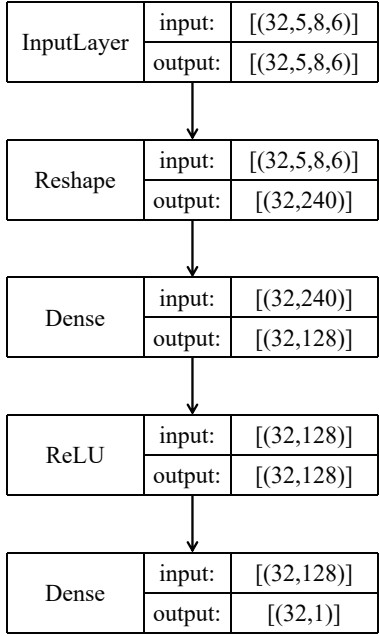

Figure 11: **MLP based model for regression**. [32,5,8,6] corresponding to the batch size, the number of gate types, the number of qubits $N$, and the depth of the circuit (refer to the width in Fig. 9), respectively.

**Fine-tune**. Recall that QuKerNet exploits two random methods in this stage to enlarge the parameter space to improve the learning performance. The first random operation refers to the number ($m \in \{1, 2 \ldots, L/L_0 - 1\}$) of reset parameterized gates in $U_E(\boldsymbol{x})$ that is random in each processing of fine-tune. And the second random operation is that the positions of reset gates are randomly selected. Combined with the above two random operations, QuKerNet can explore larger parameter space. The details of fine-tuning is shown in Fig. 10(b).

For clarification, we summarize the pseudocode of QuKerNet.

---

**Algorithm 1:** Pseudo code of QuKerNet.

---

**Input**: $\boldsymbol{x} \in \mathbb{R}^{n \times d}$, $p$, $M$, $M'$, $L_0$, $N$, $k$ and $|\Theta|$;
**Output**: $S^*$ and $\boldsymbol{\theta}^*$;

**1** Conducting feature selection on $\boldsymbol{x}$ to get $\hat{\boldsymbol{x}} \in \mathbb{R}^{\boldsymbol{n \times p}}$;
**2** Sampling $M$ circuits from $\mathcal{S}$ based on $L_0, N$, and encoding $M$ circuits to images as well as calculating KTA by Eq. (5);
**3** Train the predictor on training set of $M$ pairs (image, KTA);
**4** Sampling $M'$ circuits from $\mathcal{S}$ based on $L_0, N$, and encoding them into images ;
**5** Utilizing the predictor to predict the KTA of $M'$ circuits ;
**6** Selecting $k$ circuits with the top predicted KTA from $M'$ circuits. Fine-tuning $|\Theta|$ different $\boldsymbol{\theta}$ for each of the $k$ circuits ;
**7** To validate the test accuracy of $k|\Theta|$ kernels on $\hat{\boldsymbol{x}}$, and pick the kernel with the highest test accuracy.

---

## C  MORE SIMULATION RESULTS

In this section, we will introduce some experimental details, including the dataset, hyperparameter settings, and metric. Next, we present additional experimental results to further demonstrate the effectiveness of QuKerNet.

### C.1  DATASETS

MNIST is a dataset of handwritten digits labeled from 0 to 9, which contains 60,000 grayscale images of $28 \times 28$ resolution for training and 10,000 images for testing. While CC is a financial dataset to identify whether a credit card is fraudulent or not, *i.e.*, binary classification task, with a total of 284,807 samples (492 frauds), each of which consists of 28 features.

### C.2  HYPER-PARAMETER SETTINGS

Here, we detail the hyper-parameter settings of other classical and quantum kernels in comparison.

**RBFK**. If there is no special explanation, we set $\gamma \in \{1, 2, 3, 4, 5\}/(p \operatorname{Var}[\boldsymbol{x}_j^{(i)}])$ and $\operatorname{Var}[\boldsymbol{x}_j^{(i)}]$ is the variance of all the features $j = 1, \ldots, d$ from all the data points $\boldsymbol{x}^{(1)}, \ldots, \boldsymbol{x}^{(n)}$.

**HEAK**. In our experiments, $a \equiv X$. For the entangled gates in each block, when the index of qubit $i$ satisfies $i < N$, we apply the CNOT gate to the $i$-th and $(i+1)$-th qubits. While the $\boldsymbol{\beta}_{ij}$ corresponds to one feature in $\hat{\boldsymbol{x}}^{(i)}$.

**TEK**. We randomly initialize each $\boldsymbol{\gamma}_{ij}$ with a value uniformly sampled from the interval $[0, 2\pi)$, and set the epoch to 30 for optimization. Meanwhile, we employ gradient descent optimization with a learning rate of 0.2 to minimize the negative KTA of TEK.

Both TEK and HEAK employ a sequential encoding strategy (refer to Fig. 7(a)), with $L_0 = 1$ to encode data.

### C.3  METRIC

The Pearson correlation coefficient (PCC) is defined as:

$$\text{PCC} = \frac{\operatorname{cov}(\boldsymbol{X}, \boldsymbol{Y})}{\sigma_{\boldsymbol{X}} \sigma_{\boldsymbol{Y}}} \tag{10}$$

where $\boldsymbol{X}$ and $\boldsymbol{Y}$ are a given pair of random variables (for example, test accuracy and KTA), the cov is the covariance, $\sigma_{\boldsymbol{X}}$ and $\sigma_{\boldsymbol{Y}}$ are the standard deviation of $\boldsymbol{X}$ and $\boldsymbol{Y}$, respectively. The value of PCC ranges from -1 to 1, which implies the degree of correlation between $\boldsymbol{X}$ and $\boldsymbol{Y}$. The larger the absolute value of PCC, the stronger the correlation (-1 means they are perfectly negatively correlated, while 1 indicates they are perfectly positively correlated).

### C.4 THE TIME COMPLEXITY OF CALCULATING KTA VERSUS CALCULATING TRAIN ACCURACY

Here, we analyze the time complexity of obtaining KTA and the training accuracy of a kernel to demonstrate the computational efficiency of calculating KTA.

For a given dataset $\hat{\mathcal{D}} = \{\hat{\boldsymbol{x}}^{(i)}, y^{(i)}\}_{i=1}^{n}$, as stated in (Tsang et al., 2005), the time complexity of training and test processes of the kernelised Support Vector Machines are all $O(n^3)$, where $n$ is the number of the data. So the time complexity of calculating train accuracy is $O(n^3)$. While the time complexity of computing KTA is $O(n^2 * p)$, as there are $n * n$ entries in kernel matrix W and getting each entry needs $O(p)$ time complexity, where $p$ refers to the number of features of $\hat{\boldsymbol{x}}^{(i)}$. In general, $p \ll n$, which indicates that the computational efficiency of KTA is much higher than that of train accuracy.

### C.5 QUKERNET VERSUS RANDOM SEARCH

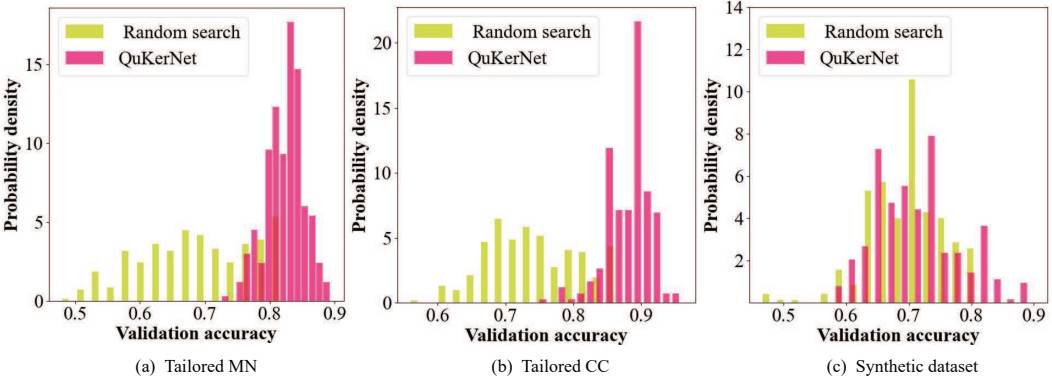

(a) Tailored MN    (b) Tailored CC    (c) Synthetic dataset

Figure 12: Validation accuracy histogram between circuit layout from random search and QuKerNet.

To further demonstrate the superiority of the QuKerNet over the random search method, we conducted a specific validation on the three datasets introduced in '**Datasets**' to evaluate the performance of the kernels based on circuit layouts searched by QuKerNet-1 and random search. In this case, $N = 8, L_0 \in \{1, 2, 3, 4, 5\}, |\mathcal{S}| = 50000, k = 200$. The performance of 200 kernels selected by QuKerNet-1 and random search are shown in Fig. 12. On tailored MNIST and tailored CC datasets, most of the kernels selected by QuKerNet-1 have better performance than the randomly searched kernels (i.e., [75%, 88%] versus [50%, 75%] on tailored MNIST). For the synthetic dataset, although the performance of the majority of kernels obtained through these two methods overlaps, there are still a few kernels that outperform those found by random search. The above findings are sufficient to demonstrate that our method is indeed capable of discovering circuit layouts that are superior to those found through random search for constructing better kernels.

### C.6 THE OPTIMAL CIRCUIT LAYOUT FOUND BY QUKERNET ON SYNTHETIC DATASET

Due to space constraints in the main body of this paper, here we provide the optimal circuit layouts obtained through QuKerNet on the systhetic dataset introduced in '**Datasets**'.

### C.7 NOISE SIMULATION RESULTS

**Performance under real device noise**. To further verify the adaptability of QuKerNet, we compared the performance of the quantum kernel searched by QuKerNet in real device noise and noiseless cases. And we employ ibmq_quito to conduct experiments, the noisy parameters are shown in Table 3. Due to the time-consuming nature of noise experiments, in order to obtain experimental results within a reasonable time, we took one-third of the synthetic data to generate new synthetic data. Furthermore, PCA is used to reduce the dimensionality of the data to 8 dimensions. In this case,

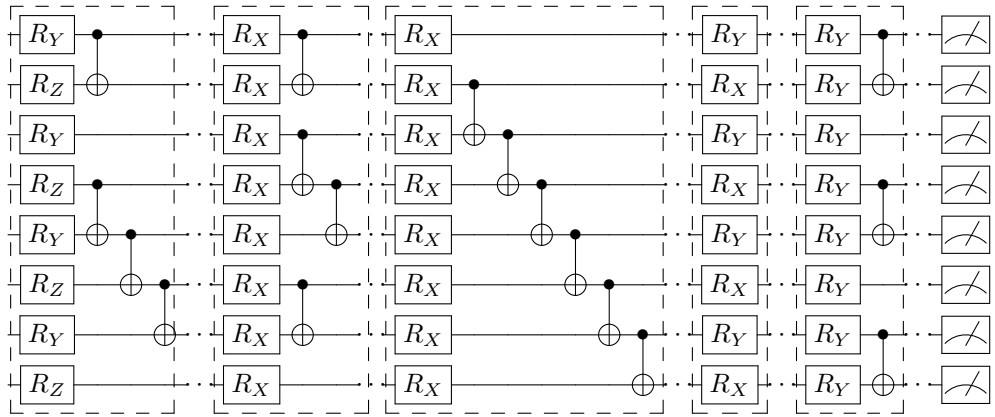

Figure 13: The layout of selected best encoding circuit for synthetic dataset (the layout of each dashed section will be repeated five times).

$M = 500, N = 4, L_0 \in \{1, 2, 3\}, |\mathcal{S}| = 20000, k = 10, |\Theta| = 10$. All experiments were run three times with different random seeds ($\gamma \in \{1, 2, 3\}/(p \text{ Var}[\boldsymbol{x}_j^{(i)}])$ for RBFK). Experimental results are shown in Fig. 14(a). Fig. 14(a) shows that the performance gaps of the kernels between the noise and noiseless cases identified by QuKerNet are not greater than 3.5% (i.e., 61.67% versus 63.33% for QuKerNet-1, 70% versus 73.33% for QuKerNet-2). Although HEAK does not show any difference in performance between the noise and noiseless conditions, its performance is significantly lower than the kernels selected by QuKerNet (i.e., 40% versus 70% in noise situations). These results indicate that QuKerNet has good adaptability to real noise conditions and can be effectively used on practical devices.

Table 3: **The noisy parameters of ibmq_quito**. $T_1$ and $T_2$ are the longitudinal and transverse relaxation time respectively. "F" represents the qubit frequency. "RE" refers to readout error of the given qubit.

| Parameter | Q1 | Q2 | Q3 | Q4 |
|---|---|---|---|---|
| $T_1(\mu s)$ | 48.21 | 60.20 | 30.10 | 70.56 |
| $T_1(\mu s)$ | 26.82 | 89.08 | 14.46 | 13.10 |
| F(GHz) | 5.30 | 5.08 | 5.32 | 5.16 |
| RE | 0.0443 | 0.0220 | 0.1111 | 0.0450 |

**Performance under different noise levels**.

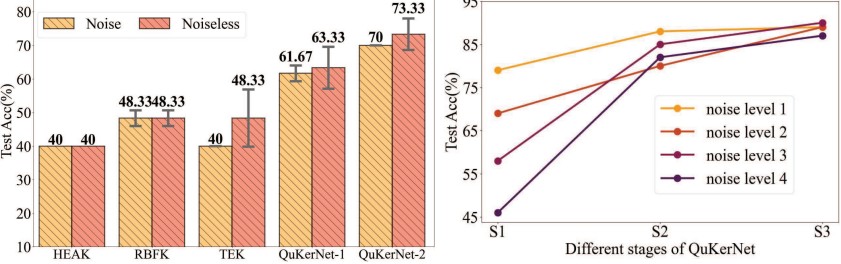

Figure 14: **Noise results of QuKerNet.** (a) The best kernels discovered by TEK, QuKerNet-1, and QuKerNet-2 on systhetic dataset in real device noise and noiseless situations. Note that the RBFK has the same results in noise and noiseless cases as it is a classical kernel method. (b) The effect of different noise levels for QuKerNet on a small part of the tailored CC dataset. S1, S2 and S3 represent the training stage, candidate stage, and fine-tuning stage, respectively. The optimal performance of kernels in each stage is shown in the line chart.

To further verify the adaptability of QuKerNet, we inspected the performance of the quantum kernel searched by QuKerNet in different noise levels. As noise simulation is time-consuming, we employ the PennyLane to simulate the noisy quantum models, and we only select 40 samples on labels 0 and 1 from the tailored CC with a 20/20 train/test split to construct the toy CC. Then we employ PCA to

reduce the dimension of toy CC to 12 dimensions. And we set up 4 noise levels are shown in Table 4. Besides, we set $M = 1500, N = 4, L_0 \in \{1, 2, 3\}, |\mathcal{S}| = 50000, k = 10, |\Theta| = 10$ to conduct our experiments. And the statistical results of five random trials are shown in Fig. 14(b). From Fig. 14(b), we know that although the performance of the kernels in training data will decrease with the increasing noise (i.e., from 79% to 46%), QuKerNet can still find kernels with high performance (i.e., around 90% test accuracy) in all noise levels, indicating that QuKerNet has better robustness and adaptability.

Table 4: The details of four noise levels (DQEP refers to depolarizing quantum error probabilities).

|  | level 1 | level 2 | level 3 | level 4 |
|---|---|---|---|---|
| DQEP of 1-qubit gate | 0.005 | 0.01 | 0.015 | 0.02 |
| DQEP of 2-qubit gate | 0.05 | 0.1 | 0.15 | 0.2 |

## C.8 ADAPTIVE CLASSICAL KERNEL VS QUKERNET

To better demonstrate the adaptability of QuKerNet, we compare the performance of Neural Kernel Network (NKN) and QuKerNet. NKN is a adaptive kernel construction method proposed in paper (Sun et al., 2018). We conducted 5 random experiments with the default parameters based on the official code of (Sun et al., 2018), and the results are summarized in the table below. The test accuracy are provided in the Table 5. The performance of NKN is lower than our method on all three datasets (i.e., 47.67% vs 94.4% on Tailored CC). Additionally, we observed that the training accuracy of NKN on the three dataset is close to 100%, indicating potential overfitting. This could be attributed to two factors: 1) We used default parameters without any optimization, and 2) The limited size of the training samples.

Table 5: The test accuracy for three datasets of different methods.

|  | RBFK | QuKerNet | NKN |
|---|---|---|---|
| Tailored MNIST | $83.60 \pm 2.52$ | $\mathbf{86.80} \pm 1.36$ | $19.73 \pm 2.78$ |
| Tailored CC | $92.80 \pm 0.40$ | $\mathbf{94.40} \pm 0.80$ | $47.67 \pm 9.23$ |
| Synthetic dataset | $71.67 \pm 2.79$ | $\mathbf{91.33} \pm 1.63$ | $50.60 \pm 5.35$ |

## C.9 ACTIVATE LEARNING

Our proposed scheme in QuKerNet offers a general framework that can be seamlessly integrated with various advanced methods and techniques. Here, we implement the method of Bayesian optimization (BO) for quantum kernel search using Optuna (Akiba et al., 2019) to show the flexibility of our approach. For the BO, We set the number of iterations to 1000, and the experimental results are shown in the Table 6. The kernel searched by Bayesian optimization did not surpass the random search approach in search space (i.e., 85.6% vs 86.8% on Tailored MNIST), which may be due to the limited number of iterations.

Table 6: BO vs random search using neural predictor.

|  | Training data | Candidate | Fine-tune |
|---|---|---|---|
| Tailored MNIST BO | $82.13 \pm 0.27$ | $83.20 \pm 1.66$ | $85.60 \pm 1.44$ |
| Tailored MNIST Ours | $82.13 \pm 0.27$ | $85.87 \pm 1.43$ | $\mathbf{86.80} \pm 1.36$ |
| Tailored CC BO | $87.00 \pm 0.00$ | $89.80 \pm 1.72$ | $93.00 \pm 1.41$ |
| Tailored CC Ours | $87.00 \pm 0.00$ | $92.20 \pm 1.17$ | $\mathbf{94.40} \pm 0.80$ |
| Synthetic dataset BO | $80.33 \pm 0.67$ | $76.67 \pm 6.15$ | $89.34 \pm 2.26$ |
| Synthetic dataset Ours | $80.33 \pm 0.67$ | $84.00 \pm 3.89$ | $\mathbf{91.33} \pm 1.63$ |

## C.10 20 QUBITS SIMULATION RESULTS

The experimental results of 20 qubits on three datasets are shown in Table 7. We set $M = 50, L_0 = 1, k = 5, |\Theta| = 5$, to condcut three random experiments. From the table, we also find that our method works well. Since our method can still search the kernels with good performance on the three datasets (e.g., from 84.89% to 86.45% on Tailored MNIST dataset).

Table 7: 20 qubits simulation results on three datasets.

|  | Training data | Candidate | Fine-tune |
|---|---|---|---|
| Tailored MNIST | $84.89 \pm 0.31$ | $86.45 \pm 0.32$ | $86.45 \pm 0.32$ |
| Tailored CC | $92.00 \pm 0.00$ | $92.20 \pm 0.00$ | $95.00 \pm 0.00$ |
| Synthetic dataset | $70.00 \pm 0.00$ | $73.33 \pm 4.71$ | $75.55 \pm 3.13$ |

## C.11 EXPERIMENTS ON HIGHER-DIMENSIONAL MNIST DATA.

To better demonstrate the effectiveness of our algorithm, we increased the dimensionality of the Tailored MNIST dataset to 80, to test our algorithm. The results in Table 8 demonstrate our method can still find kernels with better performance on higher-dimensional data.

Table 8: 80 dimension simulation results on Tailored MNIST.

| RBFK | HEAK | TEK | QuKerNet |
|---|---|---|---|
| $87.60 \pm 0.03$ | $73.33 \pm 0.00$ | $34.68 \pm 17.84$ | $\mathbf{90.27} \pm 0.53$ |

## C.12 TRADE-OFF BETWEEN THE PERFORMANCE GAIN AND OPTIMIZATION OVERHEAD.

The main trade-off between the performance gain and optimization overhead originated from finding quantum feature map with good performance within a large exponential search space.

We have conducted relevant experiments to investigate this trade-off. The main results is that, in most cases, increasing the number of fine-tuning iterations is equivalent to increase the optimization overhead, can lead to performance improvements. Detailed results are displayed in the Table 9. We observes that in most of cases when $|\Theta|$ is fixed, the performance of the selected kernel increases with an increase in $k$ (91.8% vs 93% on Tailored CC when $|\Theta|$=5). Conversely, when $k$ is fixed, the larger the value of $|\Theta|$, the better the performance of the searched kernel (84% vs 91.33% on Synthetic dataset when $k$=10).

Table 9: Trade-off results on three datasets.

|  | $|\Theta| = 0$ | $|\Theta| = 5$ | $|\Theta| = 10$ | $|\Theta| = 20$ |
|---|---|---|---|---|
| Tailored MNIST $k = 5$ | $84.53 \pm 2.44$ | $86.27 \pm 1.72$ | $86.27 \pm 1.72$ | $86.80 \pm 1.36$ |
| Tailored MNIST $k = 10$ | $85.87 \pm 1.43$ | $86.53 \pm 1.49$ | $86.53 \pm 1.49$ | $86.80 \pm 1.36$ |
| Tailored CC $k = 5$ | $91.20 \pm 1.47$ | $91.80 \pm 1.72$ | $93.20 \pm 0.75$ | $94.40 \pm 0.80$ |
| Tailored CC $k = 10$ | $92.20 \pm 1.17$ | $93.00 \pm 1.10$ | $93.40 \pm 1.02$ | $94.40 \pm 0.80$ |
| Synthetic dataset $k = 5$ | $78.67 \pm 3.86$ | $87.00 \pm 4.14$ | $89.00 \pm 1.70$ | $89.00 \pm 1.70$ |
| Synthetic dataset $k = 10$ | $84.00 \pm 3.89$ | $90.00 \pm 3.16$ | $91.33 \pm 1.63$ | $91.33 \pm 1.63$ |

