# OpenReview forum: "Neural Auto-designer for Enhanced Quantum Kernels"
_ICLR.cc/2024/Conference — ICLR 2024 poster_

### Official Review · Reviewer_3hUP · 2023-10-23

**Soundness:** 2 fair
**Presentation:** 3 good
**Contribution:** 1 poor
**Rating:** 3
**Confidence:** 5

**Summary:**

The authors propose a neural auto-designer to automatically select quantum kernels and update the internal rotation parameters. However, the problem is only the subset of well-studied Quantum Architecture Search (QAS). The proposed method is highly similar to previous literature [1] (which is not even cited in this paper). Results on three datasets are provided with the results slightly better than the selected baselines. However, the numerical experiments are extremely weak with simple datasets and naive baselines.

[1]  Quantum circuit architecture search for variational quantum algorithms.

**Strengths:**

The overall structure of this paper is ok.

**Weaknesses:**

**The authors are twisting some basic ideas and missing some critical references.**
1. The problem of quantum kernel auto-designer defined by the authors is only a special case of QAS. QAS has attracted continuous attention with numerous algorithms trying to solve this problem (see a small survey on this problem [1]). The state-of-the-art method [2] on QAS has thoroughly examined their algorithm on both VQE problems and quantum kernels (using MNIST as well). It is unclear why we should focus on only the quantum kernels instead of the general QAS problem.
2. [3] proposed a very similar QAS approach compared to this paper and [3] is not even mentioned in this paper. It is highly suspicious since the authors cite 7 other papers from the group (or more specifically from Yuxuan Du). The proposed approach is too simple with no novelty at all.
3. The experimental results are weak and lack a proper comparison. Not a single baseline from all the QAS methods is unacceptable since [2,3] and many other QAS papers all take quantum kernels as one of the testbeds for their QAS approaches. The authors constantly mentioned that the work aims to "greatly improve the power of quantum kernels under NISQ setting" but they are not considering noise models or quantum devices.


[1] QAS-Bench: Rethinking Quantum Architecture Search and A Benchmark

[2] QuantumDARTS: Differentiable Quantum Architecture Search for Variational Quantum Algorithms

[3] Quantum circuit architecture search for variational quantum algorithms.

**Questions:**

I have no further questions.

---

> ### Author Response · Authors · 2023-11-19
>
> Thanks to the precious suggestions made by the Reviewer 3hUP. These suggestions provide us with a lot of insights and help us improve the quality of our work. We are also highly grateful to the reviewer for dedicating her/his time and effort to help us improve the quality of our paper. Additionally, based on the guidance reviewer provided in the ICLR 2024, we have found that your comments do not strictly align with the criteria for a strong rejection. In response to your concerns, we have provided detailed responses and we kindly request that you reconsider the quality of our work.
>
> Q1: __*The problem of quantum kernel ...... instead of the general QAS problem.*__
>
> A1: Thanks for your comment. We will primarily address your question from the following two aspects:
> 1) __Quantum kernel search vs QAS__. Quantum kernel search is not a special case of QAS because we cannot directly apply QAS methods to design the quantum kernel due to the high complexity of evaluating the quantum kernel. Therefore, the one of main contributions of our paper is the introduction of kernel-target alignment (KTA), which greatly reduces the complexity of evaluating the quantum kernel (from $O(n^3)$ to $O(n^2p)$, $n$ is the number of samples, $p$ refers to the dimensionality of the data after dimensionality reduction), making quantum kernel search possible.
> 2) __Complementarity__. Quantum kernel search and QAS are complementary as designing a good quantum kernel. On the other hand, quantum kernel search and QAS are all can borrow ideas form Neural Architecture Search (NAS). From this perspective, quantum kernel search cannot be a subset of QAS.
>
> Q2: __*[3] proposed a very similar QAS ...... too simple with no novelty at all.*__
>
> A2: Thanks for your suggestion. We address your concerns in order.
>
> 1) __Not cite paper [3]__. Quantum kernel search is not a subset of QAS as they are complementary to each other. Additionally, our work is primarily based on paper [4], not paper [3]. Ref. [3] does not utilize a neural predictor to evaluate the performance of QNNs.
> 2) __Cite 7 other papers__. The reason we cited these seven papers is because they are highly relevant to our work. If the reviewer has concerns about any specific paper, we are open to discussing it separately. If some of them are ultimately found to be inappropriate, we will remove them accordingly.
> 3) __No novelty__. As mentioned in the first paragraph on the page 2 of our paper, “quantum kernels associated with inappropriate quantum feature maps tend to be inferior to that of classical kernels [5]. Even worse, when the quantum feature map involves deep circuit depth, a large number of qubits, and too many noisy gates, the corresponding quantum kernel may encounter the vanishing similarity [6] and the degraded generalization ability [7], precluding any potential computational advantages.” These indicates that designing a good quantum feature map is very important. However, it is not clear how to design a good quantum kernel as there is no prior knowledge to guide us in designing the kernel. Our work takes a first step in quantum kernel search. And our main innovation lies in the introduction of KTA with neural predictor to extremely reduce the time complexity of evaluating the quantum kernel, combined with feature selection to make our method applicable to high-dimensional real-world data. And our approach can be seen as a generalization in the field of quantum kernel search as we simultaneously consider the circuit layout and variational parameters.
>
> If you still have any concerns, we will provide further explanations to address them.
>
> Q3: __*The experimental results are weak ...... not considering noise models or quantum devices.*__
>
> A3: Thanks for your comment. The noise simulation results are displayed in Appendix C.7 of __our original manuscript__. The main results is that our method exhibits good adaptability across different levels of noise and performs well on real quantum devices. It is capable of finding quantum kernel with good performance.
>
> [1] Lu X, Pan K, et al. QAS-Bench: Rethinking Quantum Architecture Search and A Benchmark[J]. 2023.
>
> [2] Wu W, et al. QuantumDARTS: Differentiable Quantum Architecture Search for Variational Quantum Algorithms[J]. 2023.
>
> [3] Du Y, Huang T, et al. Quantum circuit architecture search for variational quantum algorithms[J]. npj Quantum Information, 2022, 8(1): 62.
>
> [4] Shi-Xin Zhang, et a. Neural predictor based quantum architecture search. Machine Learning: Science and Technology, 2(4):045027, 2021.
>
> [5] Hsin-Yuan Huang, et a. Power of data in quantum machine learning. Nature communications, 12(1):2631, 2021.
>
> [6] Supanut Thanasilp, et a. Exponential concentration and untrainability in quantum kernel methods. arXiv preprint arXiv:2208.11060, 2022.
>
> [7] Xinbiao Wang, et al. Towards understanding the power of quantum kernels in the nisq era. Quantum, 5:531, 2021.

---

> > ### Comment · Reviewer_3hUP · 2023-11-20
> >
> > First of all, I would like to apologize for overlooking the results in Appendix C.7. However, it seems that you are still using the simulator of Qiskit with the noise model from ibmq-quito. I would suggest the experiments with a larger scale if you are still using the noise model on a simulator since we all know the noises will scale up with the number of qubits, the number of two-qubit gates, and the circuit depth.
> >
> > Beyond that, I'm not satisfied with the answers to the three weaknesses.
> > 1. For the difference between QAS and the so-called quantum kernel auto-designer. I'm not convinced by your explanation and I hope all the reviewers including the AC should be aware of this issue. According to [1], QAS requires a given unitary matrix or a loss function to decide how to arrange different quantum gates in the circuit. For a given dataset where we use the quantum kernels to solve a specific problem, it is possible to use the loss function from the specific problem on the dataset to execute the QAS. Even if we do not have the specific problem on a given dataset, we can also use the loss function in equation 4, where we mainly consider the difference between the labels $y$. If you insist that the computation of the loss function is too time-consuming, we can also utilize KTA (which was proposed in 2001) to reduce the time complexity for QAS problems.   Therefore, we can of course utilize QAS approaches to solve the raised problem maybe with a different loss function.
> > 2. As for the similarity between this paper and [2]. The idea of using neural predictors is indeed from [3]. However, the way of dividing the search space into a layer of single-qubit gate and a layer of two-qubit gate, and generating top-k kernels, and then fine-tuning these kernels has already been proposed in  [2]. So [2] should be at least cited in this paper and including this paper will of course affect the evaluation of the novelty of this paper. Mentioning the other 7 papers from Yuxuan Du in my original review is to explain why it seems impossible to me that you have neglected this important reference since you have investigated so many papers from the group.
> > 3. As for the experiments. As long as you are using MNIST as your dataset, you should be using QAS approaches as the baselines since [2][5] and other papers use the exact same dataset and perform the exact same task in their papers (with way better performance). It actually rounds back to the problem of the difference between quantum kernel design and QAS. As far as I'm concerned, the only difference is the loss function. Once we change the loss function to the MSE loss we can surely perform the existing QAS approaches on those datasets and we are still designing quantum ansatz (which is no different than the purpose of this paper).
> >
> > To conclude, this paper utilizes existing KTA to reduce the complexity of calculating the loss, and existing neural predictor to sample ansatz to solve a problem that is very similar to QAS and can be covered by existing QAS approaches.
> >
> > [1] Lu X, Pan K, et al. QAS-Bench: Rethinking Quantum Architecture Search and A Benchmark[J]. 2023.
> >
> > [2] Du Y, Huang T, et al. Quantum circuit architecture search for variational quantum algorithms[J]. npj Quantum Information, 2022, 8(1): 62.
> >
> > [3] Shi-Xin Zhang, et a. Neural predictor based quantum architecture search. Machine Learning: Science and Technology, 2(4):045027, 2021.
> >
> > [4] Shi-Xin Zhang, Chang-Yu Hsieh, Shengyu Zhang, and Hong Yao. Differentiable quantum architecture
> > search. Quantum Science and Technology, 7(4):045023, 2022.
> >
> > [5] Wu W, et al. QuantumDARTS: Differentiable Quantum Architecture Search for Variational Quantum Algorithms[J]. 2023.

---

> > > ### Author Response · Authors · 2023-11-22
> > >
> > > Thanks to the reviewer 3hUP for providing valuable suggestions. We will now respond to them one by one.
> > >
> > > Q1: __*First of all,... and the circuit depth.*__
> > >
> > > A1: Thanks for your comment. Due to potentially long waiting times for conducting experiments using real quantum devices, we choose to perform experiments on the IBMQ-Quito backend. And the noise parameters used in this backend are indeed derived from real quantum computers, so it is sufficient to demonstrate the effectiveness of our method on a real quantum machine.
> > >
> > > Regarding larger-scale noise experiments, due to time constraints, we did not include them in the paper. However, if we obtain results in the future, we can consider including relevant findings in the appendix.
> > >
> > > To better address the review’s concern, we show the 20 qubits experimental results on three datasets for noiseless devices below.
> > >
> > > |    | __RBFK__ | QuKerNet | NKN|
> > > | -------- | -------- | -------- | -------- |
> > > | Tailored MNIST |   84.89±0.31| 	86.45±0.32	| 86.45±0.32
> > > | Tailored CC |  92.00±0.00| 	92.00±0.00| 	95.00±0.00|
> > > | Synthetic dataset| 	70.00±0.00| 	73.33±4.71| 	75.55±3.13|
> > >
> > > From the table, we also find that our method works well. Since our method can still search the kernels with good performance on the three datasets (e.g., from 84.89% to 86.45% on Tailored MNIST dataset). Before this experiment, we observed the following relationship between the number of qubits and the time required to generate a 100x100 kernel matrix: [20 qubits, 490s]; [21 qubits, 990s]; [22 qubits, 2327s]; [23 qubits, 4217s]; [24 qubits, 9112s]. This time required to compute kernel matrix is even longer in the presence of noise models.
> > >
> > > Q2: __*For the difference between QAS ... raised problem maybe with a different loss function.*__
> > >
> > > A2: Thanks for your comment. We respectfully disagree with your viewpoint that quantum kernel search is a subset of Quantum Architecture Search (QAS). Because both quantum kernel search and QAS are derivatives of Neural Architecture Search (NAS) and leverage insights from NAS to solve problems. Therefore, quantum kernel search and QAS can be considered as subfields of NAS.
> > >
> > > Besides, to better address the reviewer’s concern. We also conduct relevant experiments on the Tailored CC and Synthetic dataset based on the code provided in paper [6]. Detailed parameters are as follows, $N=5, p=5, L_0=3$, epoch=10. The results are shown in table below.
> > >
> > > |    | __Quantum kernel__ | QNN|
> > > | -------- | -------- | -------- |
> > > | Tailored CC |  66.67±0.00|	55.00±0.00|
> > > | Synthetic dataset| 59.00±0.00|	56.00±1.63|
> > >
> > > The results show that there is a significant performance gap between the quantum kernel and QNN.(e.g., 66.67% vs 55.00% on Tailored MNIST).
> > >
> > > Q3: __*As for the similarity ... many papers from the group.*__
> > >
> > > A3: Thanks for your comment. We respectfully disagree with your viewpoint that including paper [1] will affect the evaluation of the novelty of this paper. The settings of the quantum gate set, selection of top-$k$ kernels, and conducting fine-tuning are all standard practices in the field and not the novelty of our paper. Such as Refs [1-5] are all employ the neural predictor to guide the quantum circuit search, which have a very similar process to ours. Therefore, they do not affect the evaluation of the novelty of our work. The omission of reference [1] was an oversight on our part, and we will follow your suggestion to include it in the related work section for discussion.
> > >
> > > Q4: __*As for the experiments. ... quantum ansatz (which is no different than the purpose of this paper).*__
> > >
> > > A4: Thanks for your comment. We respectfully disagree with your viewpoint that quantum kernel design and QAS are only distinguished by their loss functions. As we mentioned earlier in response to your question 2 in the first round, direct optimization of quantum kernel search has high time complexity, which is why we introduced KTA. It is precisely because quantum kernel search and QAS have such fundamental differences that we did not use the QAS methods as the baselines.
> > >
> > > Regarding whether the loss function is the same or different from QAS, it is not a major focus of our paper because it does not affect the novelty of our work.
> > >
> > > To better address the reviewer’s concern. We also conducted a comparative experiment between QNN and the quantum kernel based on the paper [6]. Relevant results are displayed below.
> > >
> > > |    | __Quantum kernel__ | QNN|
> > > | -------- | -------- | -------- |
> > > | Tailored CC |  66.67±0.00|	55.00±0.00|
> > > | Synthetic dataset| 59.00±0.00|	56.00±1.63|
> > >
> > > The results from the table show that there is a significant performance gap between the quantum kernel and QNN.(e.g., 66.67% vs 55.00% on Tailored MNIST).
> > >
> > > [1] Du Y, Huang T, You S, et al. Quantum circuit architecture search for variational quantum algorithms[J]. npj Quantum Information, 2022, 8(1): 62.

---

> > > > ### Author Response · Authors · 2023-11-22
> > > >
> > > > [2] Linghu K, Qian Y, Wang R, et al. Quantum circuit architecture search on a superconducting processor[J]. arXiv preprint arXiv:2201.00934, 2022.
> > > >
> > > > [3] He Z, Zhang X, Chen C, et al. A GNN-based predictor for quantum architecture search[J]. Quantum Information Processing, 2023, 22(2): 128.
> > > >
> > > > [4] He Z, Deng M, Zheng S, et al. GSQAS: Graph Self-supervised Quantum Architecture Search[J]. arXiv preprint arXiv:2303.12381, 2023.
> > > >
> > > > [5] Deng M, He Z, Zheng S, et al. A progressive predictor-based quantum architecture search with active learning[J]. The European Physical Journal Plus, 2023, 138(10): 905.
> > > >
> > > > [6] Jerbi S, Fiderer L J, Poulsen Nautrup H, et al. Quantum machine learning beyond kernel methods[J]. Nature Communications, 2023, 14(1): 517.

---

> > > > > ### Author Response · Authors · 2023-11-22
> > > > > **Further Discussion**
> > > > >
> > > > > Dear Reviewer 3hUP,
> > > > >
> > > > > Sincere gratitude for your comment. We have addressed all the comments raised by you. Item by item responses to the your comments are listed above this response. We are looking forward to your feedback.
> > > > >
> > > > > Best wishes!
> > > > >
> > > > > Authors.

---

> > > > > > ### Author Response · Authors · 2023-11-23
> > > > > > **Any more concerns?**
> > > > > >
> > > > > > Dear Reviewer 3hUP!
> > > > > >
> > > > > > Sincere gratitude for your comment. We believe that we have addressed all the comments raised by you. Item by item responses to the your comments are listed above this response. We are looking forward to your feedback.
> > > > > >
> > > > > > Best wishes!
> > > > > >
> > > > > > Authors.

---

> > > > > > > ### Comment · Reviewer_3hUP · 2023-11-23
> > > > > > >
> > > > > > > I'm crystal clear with all the authors' claims but I would like to insist on the original evaluation.
> > > > > > >
> > > > > > > The proposed quantum kernel auto-designing problem has the same essence as the quantum architecture search problem (well-studied by the previous literature), which arranges different gates in a quantum circuit with a given loss function or a given unitary matrix. The only difference is that the quantum kernel auto-designer has a loss function that is hard to compute. Thus, the authors utilize KTA (proposed in 2001) to accelerate the computation of the loss function. The search algorithm in this paper has already been proposed by previous QAS literature. From my perspective, the contribution of this paper is at a deficient level. I would like to recommend a rejection of this paper. I understand that scoring 1 might hurt the authors' feelings so I decided to give a 3 instead.

---

> ### Author Response · Authors · 2023-11-23
>
> We would like to reiterate our deep appreciation for the reviewer's dedicated time and effort in scrutinizing our paper and providing invaluable feedback.

---

### Official Review · Reviewer_oLos · 2023-10-29

**Soundness:** 2 fair
**Presentation:** 3 good
**Contribution:** 2 fair
**Rating:** 6
**Confidence:** 5

**Summary:**

This work exploits a discrete-continuous joint optimization framework to enhance the quantum kernel, namely QuKerNet, for machine learning problems. The QuKerNet enables the automatic design of kernels by jointly both circuit layouts and variational parameters. Moreover, the authors devise a surrogate loss to efficiently optimize QuKerNet.

**Strengths:**

1. The authors put forth a QuKetNet to enhance the quantum kernel, where a joint discrete-continuous optimization framework is expected to resolve.

2. A new two-stage alternative optimization algorithm is proposed to find both the best quantum neural architecture and optimal model parameters.

3. Comprehensive numerical simulations are conducted.

**Weaknesses:**

1. Since a randomized quantum kernel has attained outstanding quantum representations and exactly corresponds to a special quantum neural network, the motivation of using an enhanced quantum kernel with automatic neural architecture search is not a key topic.

2. The computation cost of the two-stage optimization for the objective as Eq. (4) is such high that only a small subset of MNIST samples is tested for the experiments. However, the randomized quantum kernel can be simply incorporated into a quantum kernel learning architecture, which can be extended to large datasets.

3. The performance gain in a selected MNIST data subset cannot be convincing enough to demonstrate the effectiveness of the proposed enhanced quantum kernel.

4. For the empirical simulation, many hyper-parameter settings are configured, which can be particularly fitted to certain subsets but difficult to generalize to all data.

**Questions:**

1. Why does the quantum neural architecture search a key topic?

2. Since the two-stage alternative optimization has to seek a locally optimal quantum architecture, the computational cost is significantly high. So, what is the trade-off between the performance gain and optimization overhead? Is it necessary to seek an optimal quantum neural architecture?

---

> ### Author Response · Authors · 2023-11-19
>
> Thanks to the precious suggestions made by the Reviewer oLos. These suggestions provide us with a lot of insights and help us improve the quality of our work. We are also highly grateful to the reviewer for dedicating her/his time and effort to help us improve the quality of our paper. Additionally, we have provided detailed responses and hope that the reviewer will reconsider the quality of our paper.
>
> Q1:__*Since ...is not a key topic.*__
>
> A1: Thanks for your comment. As mentioned in the first paragraph on the page 2 of our paper, “quantum kernels associated with inappropriate quantum feature maps tend to be inferior to that of classical kernels [1]. Even worse, when the quantum feature map involves deep circuit depth, a large number of qubits, and too many noisy gates, the corresponding quantum kernel may encounter the vanishing similarity [2] and the degraded generalization ability [3], precluding any potential computational advantages.” These indicates that designing a good quantum feature map is very important. However, it is not clear how to design a good quantum kernel as there is no prior knowledge to guide us in designing the kernel.
>
> On the one hand, quantum kernel design is quite different from the field of deep learning, where there are many well-performing structures that can serve as basic modules to construct neural networks, such as CNN, RNN, and Transformer. In contrast, quantum machine learning is still in its early stages, and we are uncertain about which well-structured modules can be used to build quantum kernels. This situation is even more challenging in the near-term quantum devices, as it requires considering noise, topology, and various limitations. Designing a good quantum kernel becomes more challenging. In such cases, using automated search methods to design quantum kernels becomes one of the approaches to address this problem. Consequently, many researchers have conducted extensive studies in this area. In the NISQ era, quantum kernels have the potential to be a promising approach for achieving quantum advantage [4]. And quantum kernels also have a solid theoretical foundation [5-7]. Another hot topic is Quantum Architecture Search (QAS) [8-10]. There are also exists some papers related to quantum feature map design suggest that adjusting quantum kernels can improve their performance on downstream tasks [11-13].
>
> Therefore, designing better quantum kernels is both urgent and crucial, as it represents a key step towards achieving potential quantum advantage on near-term quantum devices.
>
> To address your concern, we have added a discussion on motivation of our work in the Appendix of the revised version of manuscript.
>
> Q2: __*The computation cost... to large datasets.*__
>
> A2: Thanks for your comment. We respectfully disagree with your viewpoint that the computation cost of the two-stage optimization for the objective as Eq. (4) is such high.
>
> The one of objectives of this paper is to significantly reduce the optimization complexity. Therefore, we leverage kernel-target alignment (KTA) and the neural predictor to rapidly evaluate the performance of quantum kernels. Our method has significantly reduced the time cost compared to brute force search (from $O(n^3)$ to $O(n2p)$, $n$ is the number of samples, $p$ refers to the dimensionality of the data after dimensionality reduction). More specifically, in the data collection stage, calculating the KTA is more time efficiency than computing the training accuracy on the Tailored MNIST, Tailored CC and Synthetic dataset (e.g., 288s vs 1138s, 86s vs 331s, and 48 vs 188s). While in search stage, our method is more efficient, as it can complete performance evaluation of 50,000 kernels within 75 seconds. From this point of view, the computation cost of our method would not be very high.
>
> The reason for using a small subset of MNIST samples can be attributed to two main factors. Firstly, it is widely acknowledged in the quantum machine learning to use a subset of MNIST for experimentation [1,14,15]. Secondly, this small dataset is sufficient to demonstrate the effectiveness of our method.
>
> Regarding randomized quantum kernel, we have found a reference [16]. If reviewer confirm that this is the correct paper, we can proceed with further discussion, or if reviewer provide the corresponding reference, we can discuss it accordingly.
>
> Q3: __*The performance ... quantum kernel.*__
>
> A3: Thanks for your comment. To address your concern, we have increased the dimensionality of the Tailored MNIST dataset to 80 and conducted further experiments. The results are presented in the table below. The results demonstrate our method can still find kernels with better performance on higher-dimensional data.
> |   __RBFK__ | HEAK |  TEK| QuKerNet|
> | -------- | -------- | -------- | -------- |
> | 87.60±0.03 |   73.33±0.00| 	34.68±17.84	| **90.27**±0.53

---

> > ### Author Response · Authors · 2023-11-20
> >
> > Q4: __*For the empirical simulation, many hyper-parameter settings are configured, which can be particularly fitted to certain subsets but difficult to generalize to all data.*__
> >
> > A4: Thanks for your comment. We follow the standard setting in Neural Architecture Search (NAS), as the key insight of our work is originates from NAS. On the other hand, no free lunch theory [18] also indicates that for different datasets we need different hyperparameters to get the appropriate performance. Secondly, these hyperparameters are important and commonly used in QAS [8] or NAS [17], which assist in balancing search time and performance gains. Lastly, although there are many hyperparameters in our experiments, most of them are not very sensitive. There are three hyperparameters that are worth noting, namely, $p$, $L_0$ and $N$. Among them, $p$ is the dimension after the dimension reduction of the sample. To some extent, the amount of information that the sample carrying is determined by $p$. $L_0$ and $N$ represent the number of layers and qubits, respectively. $L_0$ determines the expression ability of quantum kernel to a certain extent, and $L_0$ together with $N$ define the size of the search space.
> >
> > Q5: __*Why does the quantum neural architecture search a key topic?*__
> >
> > A5: Tanks for your comment. Please refer to the response provided for Question 1.
> >
> > Q6: __*Since the two-stage alternative optimization has to seek a locally optimal quantum architecture, the computational cost is significantly high. So, what is the trade-off between the performance gain and optimization overhead? Is it necessary to seek an optimal quantum neural architecture?*__
> >
> > A6: Thanks for your comment. We respectfully disagree with your viewpoint that the computational cost is significantly high. And the trade-off between the performance gain and optimization overhead originated from finding quantum feature map with good performance within a large exponential search space.
> >
> > One of the contributions of our paper lies in the introduction of KTA, which greatly reduces the computational cost. Please refer to the answer to question 2 regarding the computational cost. Therefore, our computational burden is not excessively high.
> >
> > About trade-off between the performance gain and optimization overhead, it is necessary to analyze the specific circumstances and make decisions accordingly. When there are numerous quantum feature maps within a search space that exhibit quantum advantage, we can quickly find a good feature map. However, if only a few quantum feature maps within the space demonstrate good performance, it may require more time to discover them. To better address the reviewer's concern, we have conducted relevant experiments to investigate this aspect. The main results is that, in most cases, increasing the number of fine-tuning iterations is equivalent to increase the optimization overhead, can lead to performance improvements. Detailed results are displayed in the table below.
> >
> > |    | __$\|\Theta\|=0$__ |  $\|\Theta\|=5$|$\|\Theta\|=10$ |$\|\Theta\|=20$|
> > | -------- | -------- | -------- | -------- |--------  |
> > | Tailored MNIST $k = 5$ |   84.53±2.44	|86.27±1.72|	86.27±1.72|	86.80±1.36
> > | Tailored MNIST $k = 10$ | 85.87±1.43|	86.53±1.49|	86.53±1.49|	86.80±1.36
> > | Tailored CC $k = 5$ | 91.20±1.47|	91.80±1.72|	93.20±0.75|	94.40±0.80
> > | Tailored CC $k = 10$ |92.20±1.17|	93.00±1.10|	93.40±1.02|	94.40±0.80
> > |Synthetic dataset $k = 5$|78.67±3.86|	87.00±4.14|	89.00±1.70|	89.00±1.70
> > |Synthetic dataset $k = 10$|84.00±3.89|	90.00±3.16|	91.33±1.63|	91.33±1.63
> >
> > To address your concern, we have added the relevant results of to the Appendix of the revised version of manuscript.
> >
> > [1] Hsin-Yuan Huang, Michael Broughton, Masoud Mohseni, Ryan Babbush, Sergio Boixo, Hartmut Neven, and Jarrod R McClean. Power of data in quantum machine learning. Nature communications, 12(1):2631, 2021.
> >
> > [2] Supanut Thanasilp, Samson Wang, Marco Cerezo, and Zoë Holmes. Exponential concentration and untrainability in quantum kernel methods. arXiv preprint arXiv:2208.11060, 2022.
> >
> > [3] Xinbiao Wang, Yuxuan Du, Yong Luo, and Dacheng Tao. Towards understanding the power of quantum kernels in the nisq era. Quantum, 5:531, 2021.
> >
> > [4] Vojtˇech Havlíˇcek, Antonio D Córcoles, Kristan Temme, Aram W Harrow, Abhinav Kandala, Jerry M Chow, and Jay M Gambetta. Supervised learning with quantum-enhanced feature spaces. Nature, 567(7747):209–212, 2019.
> >
> > [5] Maria Schuld and Nathan Killoran. Quantum machine learning in feature hilbert spaces. Physical review letters, 122(4):040504, 2019.
> >
> > [6] Schuld M, Sweke R, Meyer J J. Effect of data encoding on the expressive power of variational quantum-machine-learning models[J]. Physical Review A, 2021, 103(3): 032430.
> >
> > [7] Jonas Jäger and Roman V Krems. Universal expressiveness of variational quantum classifiers and quantum kernels for support vector machines. Nature Communications, 14(1):576, 2023.

---

> > > ### Author Response · Authors · 2023-11-20
> > >
> > > [8] Du Y, Huang T, You S, et al. Quantum circuit architecture search for variational quantum algorithms[J]. npj Quantum Information, 2022, 8(1): 62.
> > >
> > > [9] Zhimin He, Xuefen Zhang, Chuangtao Chen, Zhiming Huang, Yan Zhou, and Haozhen Situ. A gnn-based predictor for quantum architecture search. Quantum Information Processing, 22(2):128, 2023b.
> > >
> > > [10] Shi-Xin Zhang, Chang-Yu Hsieh, Shengyu Zhang, and Hong Yao. Differentiable quantum architecture search. Quantum Science and Technology, 7(4):045023, 2022.
> > >
> > > [11] Seth Lloyd, Maria Schuld, Aroosa Ijaz, Josh Izaac, and Nathan Killoran. Quantum embeddings for machine learning. arXiv preprint arXiv:2001.03622, 2020.
> > >
> > > [12] Thomas Hubregtsen, David Wierichs, Elies Gil-Fuster, Peter-Jan HS Derks, Paul K Faehrmann, and Johannes Jakob Meyer. Training quantum embedding kernels on near-term quantum computers. Physical Review A, 106(4):042431, 2022.
> > >
> > > [13] Ruslan Shaydulin and Stefan M Wild. Importance of kernel bandwidth in quantum machine learning. Physical Review A, 106(4):042407, 2022.
> > >
> > > [14] Shi-Xin Zhang, Chang-Yu Hsieh, Shengyu Zhang, and Hong Yao. Neural predictor based quantum architecture search. Machine Learning: Science and Technology, 2(4):045027, 2021.
> > >
> > > [15] Grant E, Benedetti M, Cao S, et al. Hierarchical quantum classifiers[J]. npj Quantum Information, 2018, 4(1): 65.
> > >
> > > [16] Sadowski P. Machine Learning Kernel Method from a Quantum Generative Model[J]. arXiv preprint arXiv:1907.05103, 2019.
> > >
> > > [17] Dudziak L, Chau T, Abdelfattah M, et al. Brp-nas: Prediction-based nas using gcns[J]. Advances in Neural Information Processing Systems, 2020, 33: 10480-10490.
> > >
> > > [18] Wolpert D H, Macready W G. No free lunch theorems for optimization[J]. IEEE transactions on evolutionary computation, 1997, 1(1): 67-82

---

> > > > ### Author Response · Authors · 2023-11-20
> > > > **Comment for Reviewer oLos**
> > > >
> > > > Dear Reviewer oLos:
> > > >
> > > > Thanks a lot for your efforts in reviewing this paper. We tried our best to address the mentioned concerns. Are there unclear explanations or remaining problems? We will try our best to address them.
> > > >
> > > > Kind regards,
> > > >
> > > > Authors.

---

> > > > > ### Author Response · Authors · 2023-11-21
> > > > > **Further Discussion**
> > > > >
> > > > > Dear Reviewer oLos:
> > > > >
> > > > > We really appreciate you for taking the time to review this paper. Thanks to these constructive suggestions, our paper has been largely improved. We are looking forward to hearing your further opinion on our revised paper. Thank you very much!
> > > > >
> > > > > Kind regards!
> > > > >
> > > > > Authors.

---

> > > ### Comment · Reviewer_oLos · 2023-11-23
> > > **A follow-up response to the reviewers' feedback**
> > >
> > > I thank the reviewers' feedback on my comments on this paper. The authors tried to resolve some of my concerns about the computational cost and empirical results of using neural architecture search, so I raised the score although I cannot fully agree that the NAS is a necessary component for quantum kernel learning.

---

> > > > ### Author Response · Authors · 2023-11-23
> > > >
> > > > We are delighted to see that the major concerns raised by the reviewer have been successfully addressed. We would like to reiterate our deep appreciation for the reviewer's dedicated time and effort in scrutinizing our paper and providing invaluable feedback.

---

> ### Author Response · Authors · 2023-11-22
> **Any more concerns?**
>
> Dear Reviewer oLos!
>
> Sincere gratitude for your comment. We have addressed all the comments raised by you. Item by item responses to the your comments are listed above this response. We are looking forward to your feedback.
>
> Best wishes!
>
> Authors.

---

### Official Review · Reviewer_FSEH · 2023-10-31

**Soundness:** 3 good
**Presentation:** 3 good
**Contribution:** 3 good
**Rating:** 6
**Confidence:** 4

**Summary:**

The paper proposes QuKerNet, a data-driven method to design quantum feature maps for quantum kernels. QuKerNet uses feature selection techniques to handle high-dimensional data and reduce the vanishing similarity issue. It incorporates a deep neural predictor to efficiently evaluate the performance of various candidate quantum kernels. QuKerNet jointly optimizes the circuit layout and the variational parameters of the quantum feature map, aiming to achieve the global optimal solution for the kernel design problem. QuKerNet demonstrates comparable performance over prior methods on real-world and synthetic datasets, especially for eliminating the kernel concentration issue and identifying the feature map with prediction advantages.

**Strengths:**

1. The paper is well-written and easy to follow. The paper demonstrates the superiority of QuKerNet over prior methods on various datasets, and shows the potential advantages of quantum kernels designed by QuKerNet for image/fraud classification tasks.

2. The paper employs feature-selection techniques to handle high-dimensional data and mitigate the vanishing similarity issue that affects quantum kernels on near-term quantum devices. Also the authors use a neural network to predict the performance of different candidate quantum kernels based on their circuit layouts and variational parameters, and selects the top ones for fine-tuning, which is sound and novel for enhancement of quantum kernel methods.

**Weaknesses:**

1. The author mentioned that part of the advantage of the quantum kernel comes from its being computationally hard to evaluate classically. However, the quantum kernel proposed in this paper relies on an explicit quantum feature map, meaning explicit quantum gates and quantum circuit encoding. From my view, on NISQ devices with fewer qubits and shallower quantum circuits, such a quantum feature map is not computationally hard to evaluate classically. Therefore, the reviewer believes that the quantum advantage of the proposed quantum feature map in this paper is not readily apparent. Providing a theoretical explanation rather than solely relying on empirical demonstration would make the argument more convincing.

2. In the field of quantum machine learning, optimizing both the circuit structure and the parameters of variational quantum gates for specific tasks has been previously studied. Papers [1] and [2] introduced a completely differentiable quantum architecture search (QAS) framework, avoiding the vast circuit structure search space, which the reviewer considers one of the issues faced by the proposed QuKerNet in this paper.

3. While transfer some ideas of QAS to quantum kernel methods can be considered one of the contributions of this paper, the reviewer believes the methods of QAS utilized in the paper are somewhat outdated and lack timeliness. For instance, methods based on reinforcement learning [3] and sample-free approaches [1,2] might enable the model to learn better quantum feature maps.

4. In the experimental settings of this paper, the number of utilized quantum bits seems somewhat limited (up to a maximum of 8 qubits). While the reviewer acknowledges that simulating computing overheads on a moderately sized number of qubits can be computationally intensive (even supercomputers struggle to simulate just over 30 qubits in full amplitude simulation), it is believed that the use of only 8 qubits is insufficient to demonstrate the computational gap between quantum and classical kernels. The reviewer is pleased to see more experimental results on a larger number of qubits.

[1] Wu W, Yan G, Lu X, et al. QuantumDARTS: Differentiable Quantum Architecture Search for Variational Quantum Algorithms. ICML, 2023.

[2] Lu X, Pan K, Yan G, et al. QAS-Bench: Rethinking Quantum Architecture Search and A Benchmark. ICML, 2023.

[3] Ostaszewski M, Trenkwalder L M, Masarczyk W, et al. Reinforcement learning for optimization of variational quantum circuit architectures. NeurIPS, 2021.

**Questions:**

Please see the weaknesses.

---

> ### Author Response · Authors · 2023-11-19
>
> Thanks to the precious suggestions made by the Reviewer FSEH. These suggestions provide us with a lot of insights and help us improve the quality of our work. We are also highly grateful to the reviewer for dedicating her/his time and effort to help us improve the quality of our paper. Additionally, we have provided detailed responses and hope that the reviewer will reconsider the quality of our paper.
>
> Q1: __*The author ......more convincing.*__
>
> A1: Thanks for your comment. We will separately reply your questions.
> 1) __Related to quantum feature map of NISQ devices__. We respectfully disagree with your viewpoint that our approach only suitable for NISQ devices.
>
> Before addressing your detailed concerns, we first explain our __key motivation__. As mentioned in  the first paragraph on the page 2 of our paper, “quantum kernels associated with inappropriate quantum feature maps tend to be inferior to that of classical kernels [4]. Even worse, when the quantum feature map involves deep circuit depth, a large number of qubits, and too many noisy gates, the corresponding quantum kernel may encounter the vanishing similarity [5] and the degraded generalization ability [6], precluding any potential computational advantages.” These indicates that designing a good quantum feature map is very important. On the other hand, several papers have demonstrated the advantage of quantum kernels in certain tasks or datasets. For example, Ref. [7] rigorously proves the superiority of quantum kernels in the discrete logarithm problem, while Ref. [4] demonstrates their advantage in the synthetic data. These papers indirectly confirm the potential advantage of quantum kernels. This is why our work focus on quantum kernel design.
>
> Our approach is not only applicable to NISQ devices but also to fault-tolerant quantum devices. Currently, we are in the NISQ era, where many applications are implemented on NISQ devices. Therefore, we conducted relevant experiments specifically targeting NISQ devices. We also conduct many experiments on fault-tolerant quantum devices (refers to the page 9 of our paper). Therefore, in the revised version's appendix, we will explain that our method is not only applicable to NISQ devices.
>
> Additionally, in the second sentence of the first paragraph on the page 7 of our paper, we mentioned that “can be modified to accommodate different quantum hardware platforms, enabling flexibility and adaptability.” This indicates our approach is suitable for different quantum devices, encompassing both hardware-adaptive and problem-adaptive solutions. It is not limited to NISQ devices alone. Moreover, on NISQ devices, quantum kernels can exhibit characteristics that are difficult to simulate classically, such as high entanglement or deep circuits. In such cases, the generated quantum kernels may not be easily simulated classically. On the other hand, since our approach is hardware-adaptive, as devices upgrade and the number of qubits increases, we can search for more complex kernels. These kernels are likely to be difficult to simulate classically.
>
> 2) __Quantum advantage with theoretical explanation__: Quantum advantage with theoretical explanation is challenging.
>
> We primarily aim to demonstrate from both practical and theoretical perspectives that achieving provable quantum advantage is challenging.
>
> *Practical perspective*: We hope to transfer the quantum advantage present in synthetic data to real-world data, but this is often challenging. Currently, there is no literature proposing quantum neural network (QNN) or quantum kernels with provable quantum advantage in practical problems. Firstly, quantum advantage is just one aspect that this paper focuses on. Our main emphasis is on improving the performance of quantum kernels on real-world datasets to narrow the gap with kernels on synthetic datasets. Secondly, quantum advantage is influenced by various factors. For instance, Ref. [4] demonstrates that quantum advantage is related to the structure of the data itself. Additionally, Ref. [8] provides two conditions for achieving quantum advantage, namely, quantum kernels encompasses functions that are computationally hard to evaluate classically, and the target concept resides within this class of functions. Therefore, the quantum advantage demonstrated in this paper is not necessarily obvious.
>
> *Theoretical perspective*: Paper [4] stated that *Learning* is only possible if the distribution, model and/or model selection strategy contains a lot of structure, which is not always easy to analyse theoretically. For example, deep learning can learn a distribution from data, but can not construct this distribution through mathematical formulas. This also highlights that many real-world problems cannot be easily expressed in mathematical forms, which hinders the progress of proving quantum advantage theoretically. However, problems that can be well-defined are possible candidates for demonstrating quantum advantage [4,7].

---

> ### Author Response · Authors · 2023-11-20
>
> And it is worth mentioning that proving quantum advantage theoretically is a valuable direction for future work, and it can be considered as an area of interest.
>
> Q2: __*In the field of quantum machine learning ... proposed QuKerNet in this paper.*__
>
> A2: QAS methods are __complementary__ to our approach, and our solution is not an incremental work on QAS.
>
> Thanks for your comment. QAS methods are complementary to our approach, and our solution is not an incremental work on QAS.
>
> We will address your concerns from two aspects.
> 1) __Quantum kernel vs QNN__. Quantum kernel has a rich theoretical foundation. For example, Ref. [7] rigorously proves the superiority of quantum kernels in the discrete logarithm problem, while Ref. [4] demonstrates their advantage in the synthetic data. Ref. [10] proved that kernel-based training is guaranteed to find better or equally good quantum models than variational circuit training in supervised machine learning. In contrast, there is limited research in this aspect for QNN. And quantum kernel is not equivalent to the QNN. Although there is a certain mapping relationship between quantum kernel and QNN, it is maybe that QNN requires an exponential number of qubits to achieve the same effect of the quantum kernel [11].
> 2) __QAS for quantum kernel search is not straightforward__. QAS cannot be adapted to quantum kernel search, primarily due to the requirement of calculating Kernel Target Alignment (KTA) during the evaluation process of kernel search. Obtaining KTA needs to be computed based on the kernel matrix derived from the data. As mentioned in Section 3.2 of our paper, QuKerNet is highly flexible, allowing various QAS techniques to be utilized in QuKerNet. Regarding the explanation of the differences between QAS and quantum kernel search, we will provide supplementary information and clarification in the revised version of this paper.
>
> The core idea of the two works mentioned by the reviewer [1] and [2] is consistent with Ref. [12], which is to continuousize the discrete space to accelerate the search process with advanced gradient-based methods. And in Section 3.2 of our paper, we also discuss various variants of our approach. The essence of our work lies in utilizing KTA and Neural Architecture Search (NAS) to efficiently implement quantum kernel design based on a discrete-continuous optimization problem. The reviewer's suggestion of using differentiable methods for quantum kernel search is also feasible in our framework.
>
> Q3: __*While transfer ... quantum feature maps.*__
>
> A3: Thanks for your common. We respectfully reminder the reviewer transfer some ideas of QAS to quantum kernel methods is not one of the contributions of this paper.
>
> On the one hand, QAS and quantum kernel search are all borrow ideas from NAS. On the other hand, as mentioned in our response to question 2, quantum kernels and QNNs have inherent differences, which is why we cannot directly apply the QAS approach to quantum kernel search. As mentioned earlier, the main contribution of this work is the introduction of KTA. Evaluating the performance of a kernel requires computing the kernel matrix based on the entire training data, unlike QAS which can use a smaller amount of data or fewer epochs to quickly estimate the performance of a Quantum Neural Network (QNN). Therefore, directly transferring QAS ideas to quantum kernel search is not feasible. However, by leveraging KTA, we greatly reduce the evaluation time of kernels from $O(n^3)$ to $O(n^2p)$ ($n$ is the number of samples, $p$ refers to the dimensionality of the data after dimensionality reduction), making quantum kernel search possible. More concretely, calculating the training accuracy on the three datasets (Tailored MNIST, Tailored CC, Synthetic dataset) takes 1138s, 331s, and 188s, respectively. However, the neural predictor enables us to quickly evaluate the performance of a large number of different kernels. For example, evaluating 50,000 kernels only takes 75s.
>
> We respectfully disagree with your viewpoint that the NAS method utilized in this paper is somewhat outdated and lack timeliness.
>
> Currently, in the field of quantum kernel design, there is no well-established solution. Considering the time required for kernel evaluation, we have employed the neural predictor with KTA to efficiently search kernels. This is our first step in quantum kernel search, and there is ample room for improvement. In future work, we will explore more advanced NAS approaches to design quantum kernels.
>
> The implementation method, as discussed in Section 3.2, is flexible and open to various approaches. At the same, as mentioned in our response to question 2, QAS methods are __complementary__ to our approach. Therefore, the methods mentioned by the reviewer in [1-3] are all applicable and can be considered for future work to further enhance the performance of the quantum kernel. We also discuss these approaches in revised version of this paper.

---

> > ### Author Response · Authors · 2023-11-20
> >
> > Q4: __*In the experimental settings of this paper,... The reviewer is pleased to see more experimental results on a larger number of qubits.*__
> >
> > A4: Thanks for reviewer's suggestion. We have followed the reviewer's suggestion and expanded the scale of our experiments. Although 30 qubits are simulatable, due to time constraints, we chose to conduct experiments with 20 qubits. This is based on our preliminary experiments, as we observed the following relationship between the number of qubits and the time required to generate a 100x100 kernel matrix: [20 qubits, 490s]; [21 qubits, 990s]; [22 qubits, 2327s]; [23 qubits, 4217s]; [24 qubits, 9112s]. However, we will include the results of the 30 qubits experiments in the Appendix of the paper once they are completed. The experimental results of 20 qubits are as follows:
> >
> > |    | __RBFK__ | QuKerNet | NKN|
> > | -------- | -------- | -------- | -------- |
> > | Tailored MNIST |   84.89±0.31| 	86.45±0.32	| 86.45±0.32
> > | Tailored CC |  92.00±0.00| 	92.00±0.00| 	95.00±0.00|
> > | Synthetic dataset| 	70.00±0.00| 	73.33±4.71| 	75.55±3.13|
> >
> > From the table, we also find that our method works well. Since our method can still search the kernels with good performance on the three datasets (e.g., from 84.89% to 86.45% on Tailored MNIST dataset).
> >
> > [1] Wu W, Yan G, Lu X, et al. QuantumDARTS: Differentiable Quantum Architecture Search for Variational Quantum Algorithms. ICML, 2023.
> >
> > [2] Lu X, Pan K, Yan G, et al. QAS-Bench: Rethinking Quantum Architecture Search and A Benchmark. ICML, 2023.
> >
> > [3] Ostaszewski M, Trenkwalder L M, Masarczyk W, et al. Reinforcement learning for optimization of variational quantum circuit architectures. NeurIPS, 2021.
> >
> > [4] Hsin-Yuan Huang, Michael Broughton, Masoud Mohseni, Ryan Babbush, Sergio Boixo, Hartmut Neven, and Jarrod R McClean. Power of data in quantum machine learning. Nature communications, 12(1):2631, 2021.
> >
> > [5] Supanut Thanasilp, Samson Wang, Marco Cerezo, and Zoë Holmes. Exponential concentration and untrainability in quantum kernel methods. arXiv preprint arXiv:2208.11060, 2022.
> >
> > [6] Xinbiao Wang, Yuxuan Du, Yong Luo, and Dacheng Tao. Towards understanding the power of quantum kernels in the nisq era. Quantum, 5:531, 2021.
> >
> > [7] Yunchao Liu, Srinivasan Arunachalam, and Kristan Temme. A rigorous and robust quantum speed-up in supervised machine learning. Nature Physics, 17(9):1013–1017, 2021
> >
> > [8] Jonas Kübler, Simon Buchholz, and Bernhard Schölkopf. The inductive bias of quantum kernels. Advances in Neural Information Processing Systems, 34:12661–12673, 2021.
> >
> > [9] Schuld M, Killoran N. Is quantum advantage the right goal for quantum machine learning?[J]. Prx Quantum, 2022, 3(3): 030101.
> >
> > [10] Schuld M. Supervised quantum machine learning models are kernel methods[J]. arXiv preprint arXiv:2101.11020, 2021.
> >
> > [11] Jerbi S, Fiderer L J, Poulsen Nautrup H, et al. Quantum machine learning beyond kernel methods[J]. Nature Communications, 2023, 14(1): 517.
> >
> > [12] Shi-Xin Zhang, Chang-Yu Hsieh, Shengyu Zhang, and Hong Yao. Differentiable quantum architecture search. Quantum Science and Technology, 7(4):045023, 2022.

---

> ### Comment · Reviewer_FSEH · 2023-11-20
> **Official Comment by Reviewer**
>
> Thank you for the author's reply. I believe that in some of the responses, the author fails to fully satisfy the reviewer, so I decide to maintain my score.
>
> Firstly, the quantum kernel, as a powerful tool for studying the advantages and limitations of quantum variational algorithms on real/synthetic datasets, has been theoretically investigated in related papers, including the Neural Tangents Kernel (NTK) [1] and some geometric metrics [2]. Based on this foundation, I think the author should at least adopt one of these methods to discuss whether the auto-designed quantum kernel corresponds to a quantum feature map that is stronger than manually designed ones or those generated by other QAS. The discussion should go beyond empirical observations.
>
> Furthermore, the reviewers maintain the same stance as Weakness 2. Although the quantum kernel is generally not equivalent to Quantum Neural Networks (QNN) due to the existence of some implicit quantum kernels that require an exponential number of quantum gates to implement, the reviewers believe that the proposed quantum kernel by the author still falls within the scope of QNN. The use of Kernel Target Alignment (KTA) to compute the kernel matrix and then search for the optimal kernel appears redundant. After all, the quantum circuit has already generated an explicit quantum feature map, and such a feature map can be optimized by the loss function (Eq. 4) without the immediate need to construct a kernel matrix.
>
> Lastly, it is regrettable that the author has not incorporated relevant SOTA baselines [3,4]  for comparison so far. Reviewer believes that the methods in [3,4] could potentially be strong competitors empirically surpassing the results presented in this paper.
>
> [1] Liu J, Tacchino F, Glick J R, et al. Representation learning via quantum neural tangent kernels[J]. PRX Quantum, 2022, 3(3): 030323.
>
> [2] Huang, HY., Broughton, M., Mohseni, M. et al. Power of data in quantum machine learning. Nat Commun 12, 2631 (2021).
>
> [3] Wu W, Yan G, Lu X, et al. QuantumDARTS: Differentiable Quantum Architecture Search for Variational Quantum Algorithms. ICML, 2023.
>
> [4] Lu X, Pan K, Yan G, et al. QAS-Bench: Rethinking Quantum Architecture Search and A Benchmark. ICML, 2023.

---

> ### Author Response · Authors · 2023-11-22
>
> Thanks to the reviewer FSEH for providing valuable suggestions. We will now respond to them one by one.
>
> Q1: __*Firstly, the quantum kernel,... should go beyond empirical observations.*__
>
> A1: Thanks for your comment. We kindly reminder reviewer that Quantum Neural Tangents Kernel (QNTK) [1] is used to analyse dynamic of Quantum Neural Networks (QNNs), and investigate how the QNNs converge around the global minimum. And relevant theoretical results only support on ground energy estimation [2-4]. The provable perfect classification with QNTK is still an open question. On the other hand, compared to QNNs, quantum kernel method has the advantages that it does not require gradient-descent optimization and has theoretical simplicity [5]. This is also one of the motivations for exploring the quantum kernel in this paper.
>
> Currently, the field of Quantum Architecture Search (QAS) primarily emphasizes empirical studies. For example, Refs. [6,7] employs QAS to address combinatorial optimization. Paper [8] utilizes QAS to chemistry. And these studies have made a huge difference to the quantum machine learning.
>
> However, demonstrating provable advantages of quantum kernels on real-world problems is also a sought-after goal within the field and can serve as a direction for our future work.
>
> Q2: __*Furthermore, the reviewers maintain ... immediate need to construct a kernel matrix.*__
>
> A2: Thanks for your comment. We will address your concerns one by one.
>
> 1) __Proposed quantum kernel still falls within the scope of QNN__
> We would like to kindly remind you once again that our method is adaptable to various quantum devices. When these devices undergo upgrades (e.g., the number of qubits increases, the noise decreases), the resulting kernels often become difficult to simulate classically. In such cases, the differences between quantum kernels and QNN (Quantum Neural Network) become even more pronounced.
>
> To better address the reviewer’s concern. We have conduct relevant experiments on the Tailored CC and Synthetic dataset based on the code provided in paper [11]. Detailed parameters are as follows, $N=5, p=5, L_0=3$, epoch=10. The results are shown in table below.
>
> |    | __Quantum kernel__ | QNN|
> | -------- | -------- | -------- |
> | Tailored CC |  66.67±0.00|	55.00±0.00|
> | Synthetic dataset| 59.00±0.00|	56.00±1.63|
>
> The results show that there is a significant performance gap between the quantum kernel and QNN.(e.g., 66.67% vs 55.00% on Tailored MNIST).
>
> 2) __Kernel Target Alignment (KTA) is redundant__
> We would like to kindly clarify that we did not mention using KTA to compute the kernel matrix. Instead, the computation of KTA requires the kernel matrix as a prerequisite. Our contribution lies in using KTA as a replacement for train accuracy to evaluate the performance of the kernels. This significantly reduces the time complexity of evaluating the kernels. Furthermore, as mentioned in our response to question 1, the quantum kernel does not require gradient-based optimization. This, compared to QNN, can save training time, which is another advantage. Since the training of QNN often takes an enormous amount of time and resources [12]. For example, we tested the time required for training QNN based on the code provided by the paper [11]. We set $N=5, p=5$, batch_size=64, and test on 10,000 samples and the relationship between training time and the number of parameters can be seen from the table below. We observed that the training time is directly proportional to the number of parameters. Furthermore, even with only 225 parameters (much smaller than the number of parameters in classical neural networks), training one epoch takes close to 1 hour.
>
> |  __the number of parameters__  | time |
> |:--------:| :--------:|
> | 75 |  1265s|
> | 150| 2135s|
> | 225| 3005s|
>
> The construction of the kernel matrix occurs during the data collection stage, where the kernel matrix is computed to evaluate the performance of the kernel based on KTA. This KTA value is then used as a label for training. Regarding your statement that the quantum circuit has already generated an explicit quantum feature map, but its corresponding kernel is implicit, it is important to note that evaluating the performance of a feature map alone is not sufficient as it is data-dependent. Therefore, we need to compute the kernel based on the feature map and the data in order to assess the performance of the feature map on a given dataset. Thus, computing the kernel matrix is essential.

---

> ### Author Response · Authors · 2023-11-22
>
> Q3: __*Lastly, it is regrettable ... surpassing the results presented in this paper.*__
>
> A3: Thank you for your comment. We fail to incorporate relevant SOTA baselines [9,10] for comparison, because we could not find the relevant public code. More precisely, there is no code provided in [9], and the link provided in the literature [10] is no longer accessible. Additionally, we extensively searched on platforms such as GitHub or Paper with Code but could not find the corresponding code. Furthermore, due to time constraints, we did not have enough time to reproduce the code from these papers. That is why we were unable to conduct experiments related to them. We will promptly supplement the relevant experiments, as long as the codes of Refs.[9,10] have released.
>
> [1] Liu J, Tacchino F, Glick J R, et al. Representation learning via quantum neural tangent kernels[J]. PRX Quantum, 2022, 3(3): 030323.
>
> [2] Liu J, Zhong C, Otten M, et al. Quantum Kerr learning[J]. Machine Learning: Science and Technology, 2023, 4(2): 025003.
>
> [3] Wang X, Liu J, Liu T, et al. Symmetric pruning in quantum neural networks[J]. arXiv preprint arXiv:2208.14057, 2022.
>
> [4] Nakaji K, Tezuka H, Yamamoto N. Quantum-enhanced neural networks in the neural tangent kernel framework[J]. arXiv preprint arXiv:2109.03786, 2021.
>
> [5] Shirai N, Kubo K, Mitarai K, et al. Quantum tangent kernel[J]. arXiv preprint arXiv:2111.02951, 2021.
>
> [6] Li L, Fan M, Coram M, et al. Quantum optimization with a novel Gibbs objective function and ansatz architecture search[J]. Physical Review Research, 2020, 2(2): 023074.
>
> [7] Zhang S X, Hsieh C Y, Zhang S, et al. Differentiable quantum architecture search[J]. Quantum Science and Technology, 2022, 7(4): 045023.
>
> [8] Ostaszewski M, Trenkwalder L M, Masarczyk W, et al. Reinforcement learning for optimization of variational quantum circuit architectures[J]. Advances in Neural Information Processing Systems, 2021, 34: 18182-18194.
>
> [9] Wu W, Yan G, Lu X, et al. QuantumDARTS: Differentiable Quantum Architecture Search for Variational Quantum Algorithms. ICML, 2023.
>
> [10] Lu X, Pan K, Yan G, et al. QAS-Bench: Rethinking Quantum Architecture Search and A Benchmark. ICML, 2023.
>
> [11] Jerbi S, Fiderer L J, Poulsen Nautrup H, et al. Quantum machine learning beyond kernel methods[J]. Nature Communications, 2023, 14(1): 517.
>
> [12] Zhang S X, Hsieh C Y, Zhang S, et al. Neural predictor based quantum architecture search[J]. Machine Learning: Science and Technology, 2021, 2(4): 045027.

---

> > ### Author Response · Authors · 2023-11-22
> > **Any more concerns?**
> >
> > Dear Reviewer FSEH,
> >
> > Sincere gratitude for your comment. We have addressed all the comments raised by you. Item by item responses to the your comments are listed above this response. We are looking forward to your feedback.
> >
> > Best wishes!
> >
> > Authors.

---

> > > ### Comment · Reviewer_FSEH · 2023-11-22
> > > **Official Comment by Reviewer**
> > >
> > > Thank you for further clarification. Firstly, the reviewer apologizes for the absence of code implementation for baselines [3,4] mentioned in the previous review (the reviewer found that the code had been made public but temporarily it was already taken down). The author's additional discussion and newly added experimental results further support the advantages of the proposed model, especially in numerical experiments compared to QNN (without using QAS methods). I share a similar perspective with reviewer 3hUP, that in the new revised version, the reviewer hopes the author will include baselines based on QAS methods for comparison.
> > >
> > > In conclusion, I think the authors' response has addressed my concerns, and I would like to raise the score.

---

> > > > ### Author Response · Authors · 2023-11-22
> > > >
> > > > We are delighted to see that the major concerns raised by the reviewer have been successfully addressed. And we will incorporate the baselines based on QAS methods for comparison into the final version. We would like to reiterate our deep appreciation for the reviewer's dedicated time and effort in scrutinizing our paper and providing invaluable feedback.

---

### Official Review · Reviewer_e6Tr · 2023-10-31

**Soundness:** 2 fair
**Presentation:** 3 good
**Contribution:** 2 fair
**Rating:** 6
**Confidence:** 4

**Summary:**

This paper proposes an automated pipeline to design quantum kernels. The main innovation is an optimisation algorithm that solves the joint optimisation of quantum circuit and circuit parameters that produce the quantum feature map. This uses a neural predictor for the circuit architecture performance. The method is benchmarked on toy datasets.

**Strengths:**

- Well written, tackling relevant problem of joint quantum architecture and parameter optimisation
- Novel two stage optimisation algorithm
- Neural predictor to reduce computational time, evaluation of KTA as good surrogate for accuracy
- Benchmarks on multiple datasets against multiple baselines, finding good performance

**Weaknesses:**

- The method is benchmarked against other quantum kernels and RBF. I think that a fairer benchmark would be against classical kernels that can also be adapted to data, such as [https://arxiv.org/abs/1806.04326], and choose similar number of tunable parameters. Also, I did not see how the parameters of the RBF were tuned.
- I did not see a clear benefit of the neural predictor. For each dataset one needs to train a new predictor, which requires M circuit layouts with their labels. Then M' layouts are sampled randomly and scored by the neural predictor. I assume that $M'\gg M$. It would be good to see whether sampling M layouts and scoring them with the actual labels would be worse than using the neural predictor. Also, I would expect that an active learning approach like Bayesian optimisation for the architecture search could be more beneficial since it guides the sampling rather than sampling randomly which can be very suboptimal in this large search space.

**Questions:**

- What is M' (number of sampled layouts) used in the experiments?
- Can you try against better classical baselines to give a sense of how hard these tasks are and quantify the quantum advantage?

---

> ### Author Response · Authors · 2023-11-19
>
> Thanks to the precious suggestions made by the Reviewer e6Tr. These suggestions provide us with a lot of insights and help us improve the quality of our work. We are also highly grateful to the reviewer for dedicating her/his time and effort to help us improve the quality of our paper.
>
> Q1: __*The method is benchmarked against,...... I did not see how the parameters of the RBF were tuned.*__
>
> A1: Thanks for your comment. We will address your two questions in order. The first question pertains to the issue of using a classical kernel with adaptive data, and the second question relates to the parameters of the RBF kernel.
>
> We follow the Ref. [1] and conducted 5 random experiments with the default parameters based on the official code of [1], and the results are summarized in the table below. Neural Kernel Network (NKN) is the method proposed in paper [1]. The test accuracy are provided in the table below. The performance of NKN is lower than our method on all three datasets.
>
> |    | __RBFK__ | QuKerNet | NKN|
> | -------- | -------- | -------- | -------- |
> | Tailored MNIST |   83.60±2.52| 	**86.80**±1.36	| 19.73±2.78|
> | Tailored CC |  92.80±0.40| 	**94.40**±0.80| 	47.67±9.23|
> | Synthetic dataset| 	71.67±2.79| 	**91.33**±1.63| 	50.60±5.35|
>
> We next address the concern about the parameters of the RBF. We are kindly remind the reviewer, in fact, we have introduced the relevant parameter of the RBF in the __original manuscript__. In brief, we only varied the $\gamma$ of the RBF, and $\gamma  \in \{ 1, 2, 3, 4, 5\} /(p \text{ Var}[x_j^{(i)}])$. More details are introduced in the Appendix C.2.
>
> Q2: __*I did not see a clear benefit ...... suboptimal in this large search space.*__
>
> A2: Thanks for your comment. Before addressing your detailed concerns, we first highlight our key motivation. 1) __Quantum kernel has a rich theoretical foundation__.  For example, Ref. [2] rigorously proves the superiority of quantum kernels in the discrete logarithm problem, while Ref. [3] demonstrates their advantage in the synthetic data. Ref. [4] proved that kernel-based training is guaranteed to find better or equally good quantum models than variational circuit training in supervised machine learning. In contrast, there is limited research in this aspect for Quantum Neural Network (QNN). 2) __Quantum kernel is not equivalent to the QNN__. Although there is a certain mapping relationship between quantum kernel and QNN, it is maybe that QNN requires an exponential number of qubits to achieve the same effect of the quantum kernel [5]. 3) __The potential of quantum kernel__. In the field of quantum kernel, there are two main focus. The first is the performance of quantum kernel, examining its capabilities and limitations. The second focus is whether it exhibits quantum advantage, meaning whether it can outperform classical approaches for certain tasks. Therefore, researching quantum kernels is both feasible and important.
>
> When designing quantum kernels, the only similarity with Quantum Architecture Search (QAS) is that borrowing ideas from Neural Architecture Search (NAS), transforming the design of quantum kernel into a discrete-continuous optimization problem. However, as mentioned in the last sentence of the second paragraph on the third page of our paper “The fundamental difference between neural networks and kernels hints the hardness of directly employing QAS to automatically design quantum kernels”. In order to effectively utilize NAS for quantum kernel design, we introduced Kernel Target Alignment (KTA) to reduce the time complexity (from $O(n^3)$ to $O(n^2p)$, $n$ is the number of samples,$p$ refers to the dimensionality of the data after dimensionality reduction) required for evaluating kernel performance.
>
> We next answer your questions one by one.
>
> Essentially, we are aiming to solve a discrete-continuous optimization problem. In NAS, there are many choices, and we chose to use a neural predictor because it is a mature solution in NAS [6] and has also been applied in QAS [7]. Therefore, we consider the neural predictor as our approach to explore. In the neural predictor, our main consideration is the trade-off between runtime cost and performance of quantum kernels.
>
> 1) __Benefit of neural predictor__. *Time cost*:  Evaluating the performance of quantum kernels generated by different circuits through random sampling can be extremely time-consuming. Since the time complexity of evaluating the quantum kernel with train accuracy is $O(n^3)$, and $n$ is the number of samples. For example, calculating the training accuracy on the three datasets (Tailored MNIST, Tailored CC, Synthetic dataset) takes 1138s, 331s, and 188s, respectively. Using a neural predictor can greatly reduce the time cost of evaluating kernel performance (e.g., evaluating 50,000 kernels only takes 75s). Using random sampling limits our ability to explore more kernels.

---

> ### Author Response · Authors · 2023-11-20
>
> However, the neural predictor enables us to quickly evaluate the performance of a large number of different kernels. For example, evaluating 50,000 kernels only takes 75s. *Performance*: The performance of the kernels corresponding to the randomly sampled and the ones searched through the neural predictor are shown in Fig. 12 (Appendix C.5). The main result is that the performance of kernels found by the neural predictor is significantly higher than randomly searched kernels on Tailored MNIST and Tailored CC datasets (i.e., [75%, 88%] versus [50%, 75%] test accuracy on Tailored MNIST). However, on the Synthetic dataset, the performance is slightly higher than randomly searched kernels.
>
> It is worth noting that, the neural predictor is not the only way to implement the quantum kernel search. Our goal is to automatically design a good quantum kernel, and the neural predictor is just one way to achieve that. To address your concerns better, we have included new discussions in the revised version of the paper.
>
> 2)  **Random sampling vs Neural predictor**. The reviewer’s question refers to $M$ or $M'$, so we will answer it in two steps.
>
> *If you are referring to $M$*: the validation accuracy of 200 kernels selected by QuKerNet and random search are shown in Fig. 12 (Appendix C.5). And the main result is that the performance of kernels found by the neural predictor is significantly higher than randomly searched kernels on Tailored MNIST and Tailored CC datasets, and is slightly higher than randomly searched kernels on Synthetic dataset.
>
> *If you are referring to $M'$*: the time for evaluating $M'$ kernels by randomly search is not feasible. For example, estimating 50,000 kernels on the three datasets would require approximately 56,900,000s, 16,550,000s, and 9,400,000s, respectively.
>
> 3) **Activate learning**: As mentioned in the first sentence of the first paragraph on the page 7 of our paper “Our proposed scheme in QuKerNet offers a general framework that can be seamlessly integrated with various advanced methods and techniques.”. This means we provided a paradigm that is highly flexible. And we also mentioned in the second sentence of the second paragraph on the page 2 of our paper “Specifically, ……tuning variational parameters within the quantum feature maps and employing evolutionary or *Bayesian algorithms* to continually adjust the arrangement of quantum gates within the feature map”. So we can utilize methods such as Bayesian optimization (BO) [8] or genetic algorithms [9] to guide kernel search process. These methods are complementary and compatible with our approach. Additionally, we implemented Bayesian optimization for quantum kernel search using Optuna [10] to address reviewer’s concern. We set the number of iterations to 1000, and the experimental results are shown in the table below. The kernel searched by Bayesian optimization did not surpass the random search approach, which may be due to the limited number of iterations.
> |    | __Training data__ | Candidate | Fine-tune|
> | -------- | -------- | -------- | -------- |
> | Tailored MNIST BO |   82.12±0.27|	83.20±1.66|	85.60±1.44|
> | Tailored MNIST Ours |   82.12±0.27|	85.87±1.43|	86.80±1.36|
> | Tailored CC BO|  87.00±0.00|	89.80±1.72|	93.00±1.41|
> | Tailored CC Ours|  87.00±0.00|	92.20±1.17|	94.40±0.80|
> | Synthetic dataset BO| 	80.33±0.67|	76.67±6.15|	89.34±2.26|
> | Synthetic dataset Ours| 	80.33±0.67|	84.00±3.89|	91.33±1.63|
>
> Q3: __*What is M' (number of sampled layouts) used in the experiments?*__
>
> A3: Thanks for your comment. In our experiments, $M'$ was indeed set to 50,000, which is consistent with $|S|$. This is a typo and we have made the revision in the revised version of the paper.
>
> Q4: __*Can you try ......quantify the quantum advantage?*__
>
> A4: Thanks for your suggestion. Before addressing your detailed concerns, let us first recall our statement in introduction, the second sentence of the second paragraph on the page 1, “quantum advantages may arise if two conditions are met: the reproducing kernel Hilbert space formed by quantum kernels encompasses functions that are computationally hard to evaluate classically, and the target concept resides within this class of functions.” This implies that quantum kernels with prior knowledge are likely to exhibit quantum advantage. On the other hand, we can quantify the performance gap between classical kernel and quantum kernel on a given dataset using the metric *geometric difference* proposed in reference [3]. Through this metric, we can demonstrate that quantum kernels have better performance on certain datasets that are well-suited for quantum models. That is why we chose synthetic dataset for our experiments. Additionally, we also hope to achieve quantum advantage on real-world data, which is why we selected Tailored MNIST and Tailored CC datasets.

---

> > ### Author Response · Authors · 2023-11-20
> >
> > As for the choice of the Radial Basis Function kernel (RBFK) as a baseline, it is a commonly used baseline in machine learning, which makes it a suitable choice for comparison with our proposed approach.
> >
> > To better address the concerns of the reviewer, we have also chosen the more advanced classical kernel, the Neural Tangent Kernel (NTK) [11], as an additional baseline. The experimental results of NTK with maximum depths of $d \in [1,2,3,4,5]$ are presented in the table below. It can be observed that NTK performs better than RBFK on the Tailored MNIST and Synthetic datasets (e.g., 88.54% vs 83.60%, 78.34% vs 71.67%). When compared to our approach, NTK has a slightly better performance on Tailored MNIST (e.g., 88.54% vs 86.80%). However, NTK's performance is lower than our approach on the other two datasets, which to some extent indicates that an adaptive kernel design approach for varying data is superior to a fixed kernel approach. We also found that there is a huge performance gap between NTK and QukerNet on Synthetic dataset (e.g., 78.34% vs 91.33%). This is because the Synthetic dataset is better suited for quantum models rather than classical models.
> >
> > |    | __RBFK__ | NTK| QuKerNet|
> > | -------- | -------- | -------- | -------- |
> > | Tailored MNIST |   83.60±2.52| 	**88.54**±0.27	| 86.80±1.36|
> > | Tailored CC |  92.80±0.40| 	49.96±0.00| 	**94.40**±0.80|
> > | Synthetic dataset| 	71.67±2.79| 78.34±6.67	| 	**91.33**±1.63|
> >
> > [1]Sun S, Zhang G, Wang C, et al. Differentiable compositional kernel learning for Gaussian processes[C]//International Conference on Machine Learning. PMLR, 2018: 4828-4837.
> >
> > [2] Yunchao Liu, Srinivasan Arunachalam, and Kristan Temme. A rigorous and robust quantum speed-up in supervised machine learning. Nature Physics, 17(9):1013–1017, 2021.
> >
> > [3] Hsin-Yuan Huang, Michael Broughton, Masoud Mohseni, Ryan Babbush, Sergio Boixo, Hartmut Neven, and Jarrod R McClean. Power of data in quantum machine learning. Nature communications, 12(1):2631, 2021.
> >
> > [4] Schuld M. Supervised quantum machine learning models are kernel methods[J]. arXiv preprint arXiv:2101.11020, 2021.
> >
> > [5] Jerbi S, Fiderer L J, Poulsen Nautrup H, et al. Quantum machine learning beyond kernel methods[J]. Nature Communications, 2023, 14(1): 517.
> >
> > [6] Wen W, Liu H, Chen Y, et al. Neural predictor for neural architecture search[C]//European Conference on computer vision. Cham: Springer International Publishing, 2020: 660-676.
> >
> > [7] Shi-Xin Zhang, Chang-Yu Hsieh, Shengyu Zhang, and Hong Yao. Neural predictor based quantum architecture search. Machine Learning: Science and Technology, 2(4):045027, 2021.
> >
> > [8] Duong T, Truong S T, Tam M, et al. Quantum neural architecture search with quantum circuits metric and bayesian optimization[J]. arXiv preprint arXiv:2206.14115, 2022.
> >
> > [9] Leo Sünkel, Darya Martyniuk, Denny Mattern, Johannes Jung, and Adrian Paschke. Ga4qco: Genetic algorithm for quantum circuit optimization. arXiv preprint arXiv:2302.01303, 2023.
> >
> > [10] Akiba T, Sano S, Yanase T, et al. Optuna: A next-generation hyperparameter optimization framework[C]//Proceedings of the 25th ACM SIGKDD international conference on knowledge discovery & data mining. 2019: 2623-2631.
> >
> > [11] Arora S, Du S S, Li Z, et al. Harnessing the power of infinitely wide deep nets on small-data tasks[J]. arXiv preprint arXiv:1910.01663, 2019.

---

> > > ### Author Response · Authors · 2023-11-20
> > > **Comment for Reviewer e6Tr**
> > >
> > > Dear Reviewer e6Tr:
> > >
> > > Thanks a lot for your efforts in reviewing this paper. We tried our best to address the mentioned concerns. Are there unclear explanations or remaining problems? We will try our best to address them.
> > >
> > > Kind regards,
> > >
> > > Authors.

---

> > ### Comment · Reviewer_e6Tr · 2023-11-20
> >
> > Thank you for your response. I do not have further comments.

---

> > > ### Author Response · Authors · 2023-11-20
> > >
> > > Thank you for your reply. Have a good day!

---

### Meta-Review · Area_Chair_6TVN · 2023-12-02

**Metareview:**

The paper propose a novel way QuKerNet to design quantum kernels from learning feature maps in a data-adaptive manner. Base on a  two stage optimization procedure, QuKerNet utilizes input dataset, reduces in computational time, and improves performances on benchmark dataset; and I believe it would be of interest of the ICLR audience and the community. From the fruitful discussions, most of the reviewer's concerns have be resolved and I would strongly recommend the authors integrate the discussion points into the revised manuscript to further improve the paper.

**Justification For Why Not Higher Score:**

The innovation from previous literature, as well related to QAS can be a bit limited.

**Justification For Why Not Lower Score:**

The paper is generally well-written, and the results could be of interest to the community. 3/4 reviewers stated their concerns have been clarified and rated the paper above the acceptance threshold. Despite strong rejection view from one reviewer, which is mainly based on novelty of the method and literature review, I may think it may still be beneficial for the community to see this work in the conference.

---

### Decision · Program_Chairs · 2024-01-16

Accept (poster)